# Unlocking carrier confluence in covalent organic frameworks for efficient photoreduction of dilute nitrate to ammonia

Yang Su[1,2], Zhe Wang[1], Xiaoxu Deng[2] ✉, Shuang-Feng Yin [3,4] ✉ & Peng Chen [1,2] ✉

Photocatalytic reduction of nitrate to ammonia is a promising route for sustainable nitrogen recycling, but its efficiency is often limited by disordered charge migration, interlayer charge depletion, and insufficient reactant activation, especially under dilute conditions. To address these challenges, an asymmetric spatial polarity strategy is applied to regulate polar distribution in donor-acceptor covalent organic frameworks at both molecular and layered levels. Strong intramolecular polarity confines charge transfer pathways, while convergent interlayer polarity enhances the internal electric field and promotes directional charge migration. Differentiated polar active sites facilitate nitrogen-oxygen bond cleavage, hydrogen intermediate formation, and nitrate activation in water. Here, we show that the optimized photocatalyst achieves an ammonium production rate of 0.758 mmol g$^{-1}$ h$^{-1}$ and an areal activity of 20.363 mmol cm$^{-2}$ under natural sunlight, demonstrating competitive performance for nitrate reduction under dilute conditions.

$NO_3^-$ is widely present in industrial wastewater, agricultural runoff, and groundwater, presenting a threat to human health[1–4]. Among the reported technologies for nitrate removal, the photocatalytic nitrate reduction to ammonia (PNRA) driven has demonstrated a win-win strategy for energy conversion and environmental protection[2,5]. Although PNRA has achieved significant progress in basic research, most of the reported catalysts are operated under high $NO_3^-$ concentration ($\geq$ 100 mM) conditions, which requires additional consideration of the high cost associated with concentrating the $NO_3^-$ solution[6–8]. However, real-world situations frequently involve polluted streams with low $NO_3^-$ concentrations, such as industry wastewater (<50 mM) and contaminated groundwater (<2 mM)[9–14]. However, the efficient conversion of low-concentration $NO_3^-$ is fraught with challenges, which include a complex eight-electron-nine-proton transfer process, limited low-concentration adsorption and enrichment, as well as low carrier transport efficiency[15,16].

Consequently, developing photocatalysts with high activity, selectivity, and stability for the reduction of low-concentration $NO_3^-$ to $NH_3$ is of great significance.

In principle, nitrate reduction fundamentally is a deoxygenation and hydrogenation process[17–20]. However, they face two major challenges: the low surface coverage of dilute nitrate limits access to active sites, and the kinetically challenging water dissociation step restricts the supply of crucial active hydrogen (*H)[14,21,22]. Therefore, excessive adsorption of nitrate intermediate induces side reactions, while the hydrogen-rich adsorbed state promotes the reduction of nitrate to produce hydrogen[23–25]. Therefore, it is of paramount importance to disrupt the equilibrium relationship between the adsorption of nitric acid and active hydrogen.

Recently, covalent organic frameworks (COFs) have attracted substantial attention for nitric acid reduction due to their high crystallinity, stable structures, numerous pore channels, and abundant

[1]Guizhou Provincial Key Laboratory of Green Catalysis and Materials for Resource Conversion, School of Chemistry and Chemical Engineering, Guizhou University, Guiyang, Guizhou, China. [2]College of Big Data and Information Engineering, Guizhou University, Guiyang, Guizhou, China. [3]College of Chemistry and Chemical Engineering, Hunan University of Science and Technology, Xiangtan, P. R. China. [4]Advanced Catalytic Engineering Research Center of the Ministry of Education, State Key Laboratory of Chemo/Biosensing and Chemometrics, College of Chemistry and Chemical Engineering, Hunan University, Changsha, P. R. China. ✉e-mail: dxjdeng7@163.com; sf_yin@hnu.edu.cn; pchen3@gzu.edu.cn

functional groups[12,26–29]. Furthermore, these functional groups establish diverse active sites that not only enhance the adsorption of nitric acid and stabilize key intermediates, but also facilitate the continuous supply of active hydrogen, thereby achieving synergistic control over the intermediate stages of hydrogenation and dehydrogenation pathways[30]. Most works adopting the A-D-A configuration in COFs have demonstrated that such donor-acceptor-donor frameworks can, in principle, provide a huge built-in electric field and promote charge separation within the structure[31,32]. However, this configuration also exhibits intrinsic limitations when excessive dual reduction centers and irregular stacking occur. The uncontrolled dual charge migration pathways may induce charge-transfer disorder, amplify Coulomb interactions, and enhance dielectric screening, collectively weakening crystallinity and accelerating charge recombination before carriers reach reactive surface sites[33]. More importantly, owing to the interlayer interactions, there exists an irreconcilable contradiction in the direction of carrier migration between the interlayer carriers and the in-plane carriers. Thus, elucidating carrier transport across and within

multilayer structures remains a critical yet poorly understood impediment to advancement.

Herein, we present a revolutionary asymmetric spatial polarity strategy that enables precise control of polar distribution both within the intramolecular and across the layers in donor-acceptor COFs. This approach fundamentally addresses long-standing challenges in carrier management and reactant activation for PNRA under ultralilute conditions—without the need for cocatalysts or sacrificial agents. Our findings uncover several key mechanistic insights: 1) Incorporating structurally distinct polar groups as linkers endows the COF layers with in-plane strong polarity, which constrains charge transfer pathways and mitigates the inherent risk of carrier dissipation in multidirectional systems. 2) Weakened interlayer coupling, together with spatially convergent polarity, promotes the formation of polar channels and unconventional longitudinal polarization, strengthening the internal electric field and enabling directional migration of interlayer charges. These features collectively enhance charge transport both within and between layers (Fig. 1a). 3) Establishing differentiated bipolar sites not

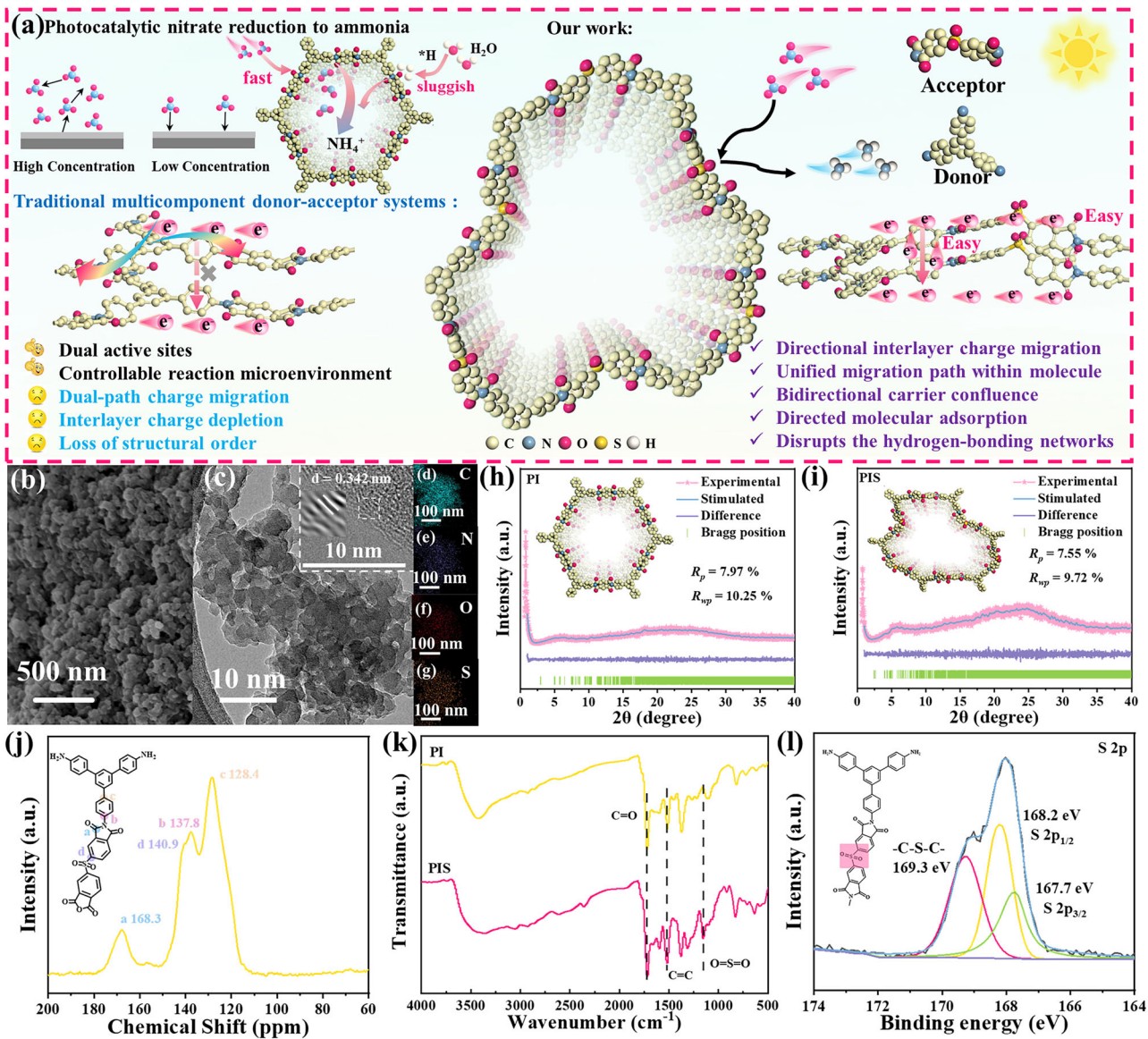

**Fig. 1 | Synthesis and structural characterization of photocatalysts. a** Schematic diagram of carrier migration in PIS (PI without the following advantages: dual-path charge migration, interlayer charge depletion, and loss of structural order). **b** SEM image, (**c**) TEM image (sketch: HR-TEM image of PIS), and (**d–g**) elemental mapping images of PIS. Experimental and simulated PXRD patterns of (**h**) PI and (**i**) PIS. **j** Solid-state $^{13}$C NMR of PIS. **k** FITR spectrum of all samples. **l** XPS spectra of S 2$p$ for PIS.

only facilitates the cleavage of the N-O bond and the generation of *H but also disrupts the hydrogen-bonding networks in water and promotes the adsorption and diffusion of trace nitrate ions. As a result, the optimized photocatalyst achieves a highly competitive $NH_4^+$ production rate of 0.758 mmol $g^{-1}$ $h^{-1}$ in ultra-dilute nitrate solution (0.99 mM). Moreover, under 10 h of natural sunlight exposure, it also demonstrates considerable activity (20.363 mmol $m^{-2}$). This work not only fills a critical knowledge gap in manipulating interlayer charge transfer via polarization engineering but also provides deep insights into enhancing photocatalytic efficiency under low-concentration conditions.

## Results

### Structure and morphology characterizations

The synthesis route involves the connection of 1,3,5-tris(4-aminophenyl) benzene (TAPB) with pyromellitic dianhydride and 5,5'-sulfonylbis(isobenzofuran-1,3-dione) through a rapid and straightforward polycondensation method, which can be donated as polyimide (PI) and polyimide-sulfonyl (PIS), respectively (Supplementary Fig. 1). The scanning electron microscopy (SEM) images in Fig. 1b and Supplementary Fig. 2 reveal that both PI and PIS exhibit coral-like spherical morphologies. High-resolution transmission electron microscopy (HR-TEM) observations reveal a characteristic lattice fringe spacing of 0.342 nm in PIS, which corresponds to the interlayer stacking resulting from π-π interactions (Fig. 1c)[34]. In addition, a uniform distribution of C, N, O and S elements was observed in PIS (Fig. 1d–g). $N_2$ adsorption was used to characterize the surface area and porosity of PIS and PI. PIS exhibits a much higher surface area (41.58 $m^2$ $g^{-1}$) than PI (10.95 $m^2$ $g^{-1}$) (Supplementary Fig. 3). Their pore sizes are about 2.08 and 1.93 nm, respectively, close to the theoretical value of 2.0 nm and consistent with the structural design.

As stated in Supplementary Fig. 4, the characteristic diffraction peak in the X-ray diffraction (XRD) pattern at 24.09° corresponds to the (001) lattice plane. This can be attributed to the π-π stacking distance between neighboring molecules[35]. Notably, the peak intensities and positions are in good agreement with the simulated powder X-ray diffraction (PXRD) patterns of the AA stacking modes. Pawley refinement revealed negligible differences (Fig. 1h, i, gray curves) from the experimental results, showing small $R_{wp}$ and $R_p$ values of 9.72% and 7.55% for PIS, 10.25% and 7.97% for PI, respectively. As shown in Fig. 1j, the solid-state $^{13}C$ nuclear magnetic resonance (NMR)

Analysis of PIS shows that the signals at 137.8 and 128.4 ppm belong to C-N and C = C bonds[36], while the peak at 140.9 ppm is associated with the C-S bond[37]. Additionally, $^{13}C$ NMR further confirmed the formation of the imide linkage with the characteristic carbonyl carbon of the imide ring appearing at around 168.3 ppm[38]. Besides, the rest of the peaks located in the range of 100-140 ppm corresponded to the highly conjugated phenyl framework[39]. In addition, the bonding state was further confirmed by Fourier transform infrared spectroscopy (FTIR) (Fig. 1k). Characteristic peaks at 833.2 $cm^{-1}$ (aromatic C-H), 1519.5 $cm^{-1}$ (aromatic C = C), and 1723.2 $cm^{-1}$ (carbonyl C = O) were identified in both PIS and PI particles[40,41]. Notably, the peak at 1152.2 $cm^{-1}$ (O = S = O stretching vibration) was only observed in PIS particles due to the incorporation of 5,5'-sulfonylbis(1,3-isobenzofurandione)[42,43].

To gain deeper insight into the bonding characteristics of the catalysts, all samples were analyzed using X-ray photoelectron spectroscopy (XPS). As shown in Supplementary Fig. 5a, the C 1s spectrum of PIS was deconvoluted into three peaks at 284.8, 285.6, 286.2, and 289.1 eV, corresponding to the formation of C = C, C-N, C-S, and C = O bonds[44–46]. In the O 1s spectrum, three fitted peaks at 533.4, 532.2, and 531.6 eV were assigned to S=O, surface-adsorbed oxygen, and N-C=O in PIS (Supplementary Fig. 5b)[40,47]. Furthermore, the S 2p spectra were deconvoluted into three peaks in Fig. 1l, including 167.7 and 168.2 eV that correspond to S $2p_{3/2}$ and S $2p_{1/2}$, as well as 169.3 eV that fitted with C-S-C[43,48]. As shown in Supplementary Fig. 5c, the N 1s spectrum

exhibits a peak at 400.2 eV, which originates from the C-N = O bond[19,49]. Notably, the PI sample exhibits a similar structural composition. Taken together, the results of structural characterizations verify that the prepared samples have the anticipated structure.

### Directional modulation of interlayer charges

The optical properties of PIS and PI were characterized by UV-vis diffuse reflectance spectroscopy. As shown in Supplementary Fig. 6a, the light-harvesting capability of PIS was significantly improved and extended across the visible light range by incorporating 5,5'-sulfonylbis (isobenzofuran-1,3-dione) as a building block. The optical band gaps of PIS and PI were deduced to be 1.67 and 1.50 eV, respectively, from Tauc plots (Supplementary Fig. 6b)[50]. Moreover, the conduction band (CB) positions of PI and PIS were determined to be −0.25 and −0.32 V, respectively (Supplementary Fig. 7). Consequently, the valence band (VB) edge positions of PI and PIS were calculated to be 1.25 and 1.35 V, respectively. As shown in the energy band structure, the conduction band of PIS is higher than the reduction potential of $NO_3^-$, indicating that these structures are favorable for the photocatalytic conversion of $NO_3^-$ into ammonia (Supplementary Fig. 8). A detailed examination of the electronic band structure was conducted through density functional theory (DFT) calculations, focusing on the frontier molecular orbitals. As displayed in Supplementary Fig. 9, the Highest Occupied Molecular Orbital (HOMO) of PIS is located predominantly on TAPB, while the Lowest Unoccupied Molecular Orbital (LUMO) is centered on 5,5'-sulfonylbis(isobenzofuran-1,3-dione), indicating a distinct donor-acceptor (D-A) structure.

Although the prepared samples exhibit donor-acceptor (D-A) characteristics, most exhibit distinct geometric conformations and molecular polarities. Predominantly, the challenge of overcoming the issue of charge recombination resulting from structural heterogeneity persists as a significant hurdle. As shown in Supplementary Fig. 10, for PI, more positive charges are generated on the TAPB unit, while substantial negative charges accumulate on the pyromellitic dianhydride (PMDA) moiety. For PIS, an increased density of positive charges is observed on TAPB, while a substantial number of negative charges accumulates on C = O and O = S = O groups. Notably, the charge of the O = S = O group is more negative than that of the C = O group. Since the differentiated polar centers lead to a strong concentration of electric charges, there is an uneven spatial distribution of charges, thus generating an intramolecular local dipole. Therefore, the dipole moment of PIS is 7.9776 D, which is significantly larger than that of PI (6.5974 D), suggesting the considerably high charge migration driving force in the plane. This result suggests that the differentiated polar functional groups have enhanced in-plane polarization and facilitated the migration of intranodal charge carriers.

Although the D-A structures facilitate the transmission of electric charges in a plane, achieving the directional regulation of interlayer charges depends on a highly intrinsically stacked structure. Consequently, the charge distribution and the difference between the ground- and excited-state dipole moments ($\Delta\mu_{ge}$) of the double-layer structure were calculated and investigated. As shown in Fig. 2a, PIS exhibits relatively large charge moments of −2×$10^{-5}$, −4×$10^{-5}$, and 6×$10^{-5}$ eV/Å in the x, y, and z directions, respectively, which are greater than those of PI (1×$10^{-5}$, −1×$10^{-5}$, and 1×$10^{-5}$ eV/Å). Notably, the charges in z-directions in PIS are higher than those in PI. Furthermore, the charges of the O = S = O groups in the PIS of the upper layer are significantly higher than those in the lower layer. Conversely, there is only a minor difference in the charges of the C = O bonds (Supplementary Fig. 11). In PI, the charge difference of the C = O group is not substantial (Supplementary Fig. 12). This reason can be elucidated as follows: PIS, featuring a prominent vertical structure and planar charge aggregation function, disrupts the linear structure of PI, thereby facilitating the three-dimensional polarization of charges. However, in the bilayer structure of PIS, strong interlayer π-π interactions induce significant

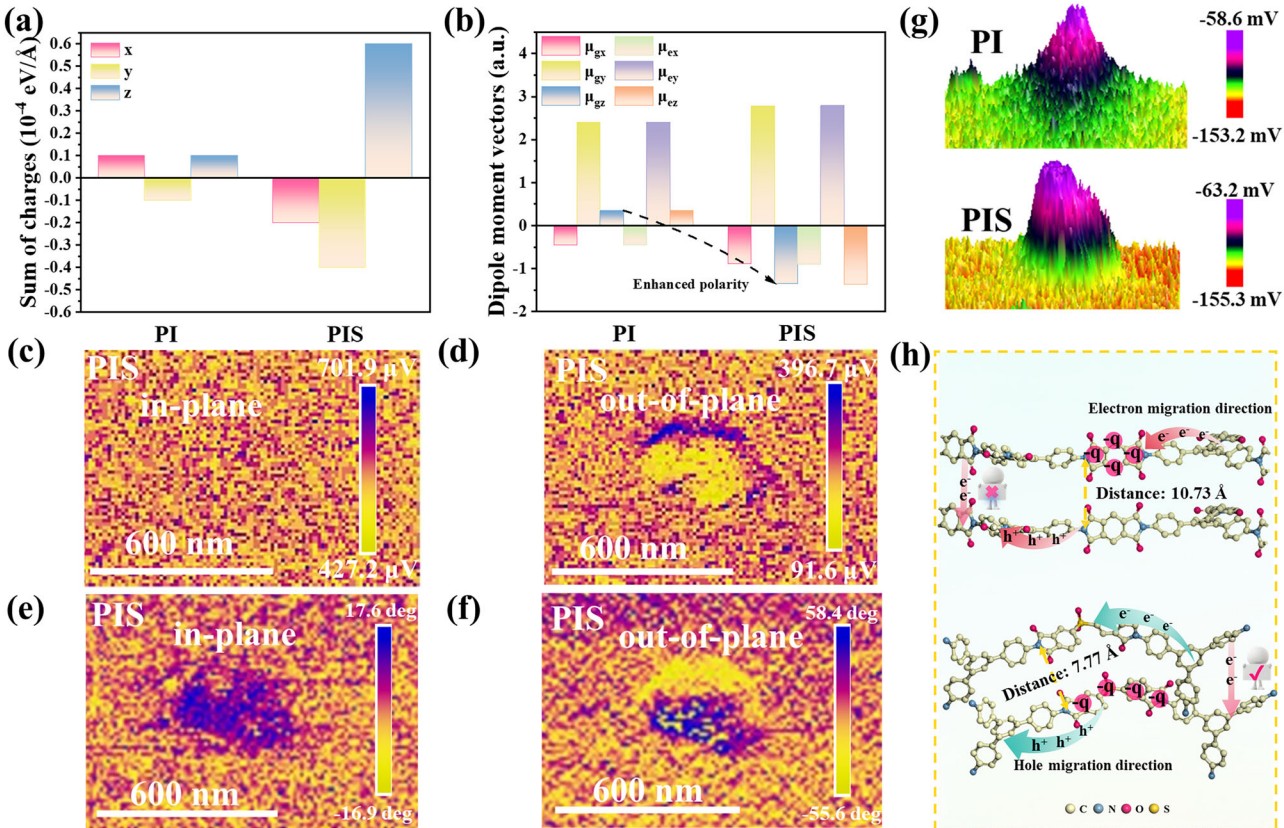

**Fig. 2 | Research on the Mechanism of Carrier Separation Induced by Dipole Fields. a** The sum of the charges of all samples in all directions. **b** The vector of ground- and excited-state dipole moments for all samples. Piezoresponse force microscopy (PFM) of PIS: **c** In-plane amplitude image, (**d**) out-of-plane amplitude image, (**e**) in-plane phase image and (**f**) out-of-plane phase image. **g** Surface potential with a KPFM of PI and PIS. **h** Charge transfer mechanism diagram.

overlap of electron clouds between adjacent layers. Such interlayer electron cloud overlap can form a "polar channel", facilitating interlayer carrier hopping. Furthermore, the enhanced interlayer interactions may optimize the intralayer atomic arrangement through stress transfer, thereby indirectly improving the intralayer charge mobility (Supplementary Fig. 13). This results in a stronger local electric field, with further enrichment of electron density on the negatively charged side within each layer and a more negative atomic potential. As shown in Fig. 2b, the dipole moment components ($\mu_{gx}$, $\mu_{gy}$ and $\mu_{gz}$) of PIS (−0.89, 2.78, and −1.35) along the x-, y-, and z-directions are significantly larger than those of PI (−0.45, 2.40, and 0.34), suggesting PIS exhibits significantly stronger polarization in its three-dimensional structure. Moreover, it also exhibits highly directional polarity in the excited state, indicating that it overcomes the variable spatial configurations that may allow for electron back-diffusion during migration (Supplementary Table 1). Consequently, it shows higher macroscopic polarity in the ground state and the excited state. Above all, the O = S = O groups in the material enable vectorial superposition of their dipole moments, forming an unconventional longitudinal polarization and "polar channels" that direct carriers along low-resistance paths while suppressing reverse movement.

To verify the longitudinal polarization, Piezoresponse force microscopy (PFM) was employed. As presented in Fig. 2c–f and Supplementary Fig. 14, PIS exhibits not only in-plane characteristics but also abnormal out-of-plane polarization phenomena along the non-polarized crystallographic orientation, whereas PI only demonstrates in-plane polarization characteristics. Additionally, the in-plane phase signal of PIS is significantly lower than its out-of-plane counterpart, indicating that its polarization is predominantly oriented in the longitudinal polarization. This phenomenon can be attributed to the

asymmetrically distributed O = S = O polar groups, inducing the non-uniform charge distribution within the molecular plane, thereby strengthening the in-plane dipole vectors[51]. In addition, the spatially configured S = O group disrupts the polarity distribution within the plane, enabling vectorial superposition of their dipole moments, forming an unconventional longitudinal polarization and "polar channels" that direct carriers along low-resistance paths while suppressing reverse movement. Therefore, it will form a huge IEF to facilitate the migration of charge carriers. As shown in Fig. 2g (Kelvin Probe Force Microscopy, KPFM), the surface potentials of PIS and PI are 22.80 mV and 45.80 mV, respectively. Moreover, the zeta potentials of PIS and PI are recorded as −10.15 mV and −3.67 mV, respectively (Supplementary Fig. 15 and Supplementary Table 2). Therefore, the IEF of PIS is 2.36 times stronger than that of PI (Supplementary Fig. 16), indicating that the O = S = O moiety possesses greater polarity, which is consistent with the polarization-electric field (P-E) results (Supplementary Figs. 17 and 18).

Therefore, the asymmetrically distributed O = S = O polar groups induce an uneven charge distribution within the molecule, thus enhancing the in-plane dipole vector. Simultaneously, its spatial configuration disrupts the planar polarity distribution, enabling the superposition of dipole moment vectors. This leads to the formation of a longitudinal polarization and a "polar channel" that guides the directional transport of charge carriers and suppresses electron recombination.

## Carrier mobility efficiency

To further investigate the charge migration trend, in-situ XPS analysis was conducted to explore the photogenerated electron redistribution among these dipole centers. As illustrated in Supplementary

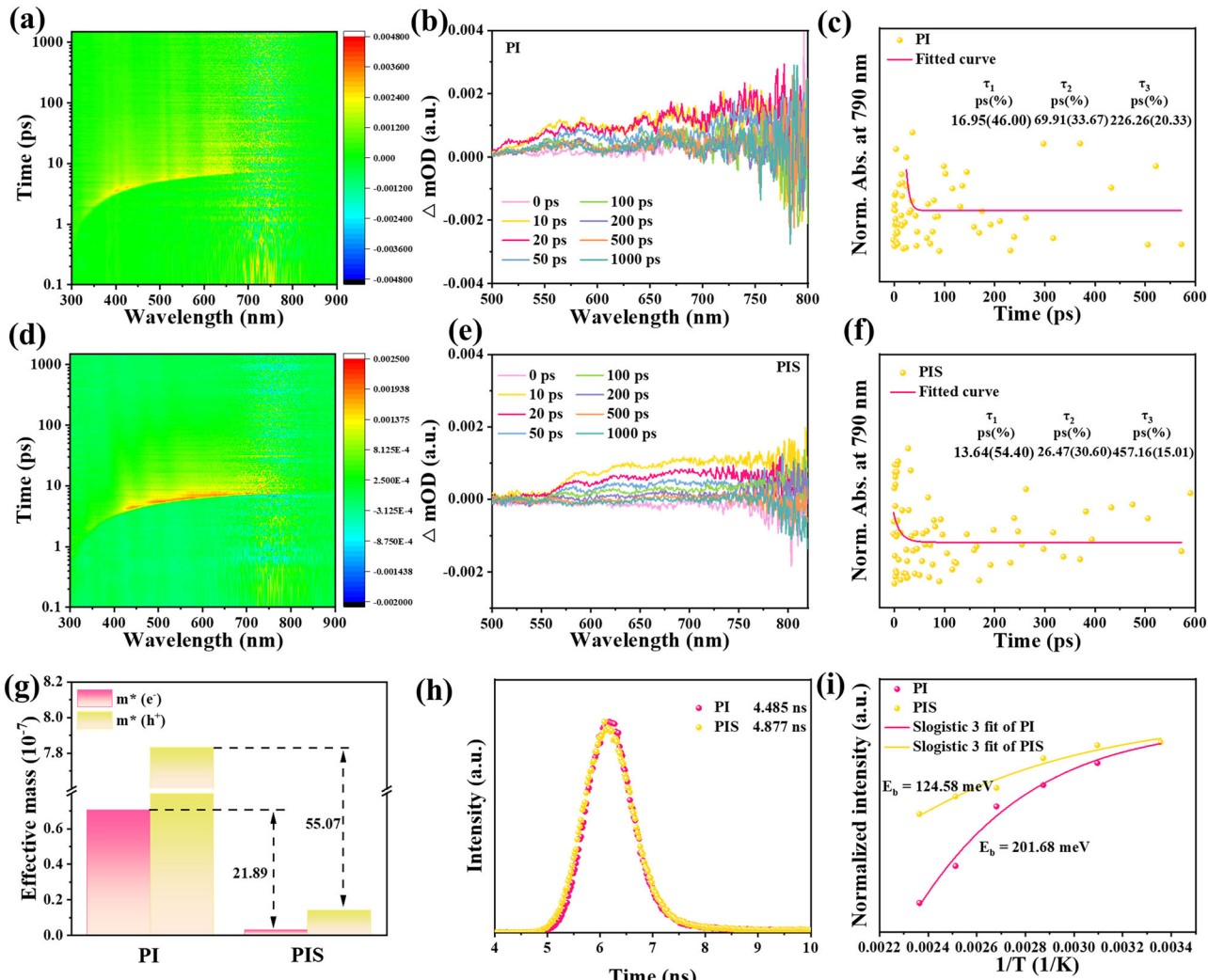

**Fig. 3 | Research on carrier migration in PIS. a** 2D mapping of femtosecond transient absorption (fs-TA) spectroscopy for PI. **b** Transient absorption spectra of PI. **c** Kinetics of the characteristic fs-TA absorption bands at 790 nm for PI. **d** 2D mapping of femtosecond transient absorption (fs-TA) spectroscopy for PIS. **e** Transient absorption spectra of PIS. **f** Kinetics of the characteristic fs-TA absorption bands at 790 nm for PIS. **g** Electron and hole effective masses in the presence of SE coupling. **h** Time-resolved fluoroimmunoassay of the as-prepared samples. **i** Temperature-dependent PL spectra of as prepared samples as a function of temperature.

Figs. 19 and 20, the binding energies of the C = O and S = O groups in O $1s$ spectra and S $2p$ spectra for PIS shift to lower values under light irradiation, indicating electron depletion and confirming that the dual dipole centers undergo charge redistribution upon illumination. In contrast, the C = O groups in PI exhibit negligible changes, suggesting the weak electron driving force of the D-A structure. The photogenerated charge carrier dynamics were elucidated through femtosecond transient absorption spectroscopy (fs-TA). As depicted in Fig. 3a, d, the red-colored region denotes the excited-state absorption (ESA) signal, whereas the blue-colored region signifies the ground-state bleaching (GSB) signal. As shown in Fig. 3b, e, following 350 nm photoexcitation, a broad positive absorption feature spanning 500–700 nm was observed and attributed to excited-state absorption (ESA). The resolved spectral peaks confirm that ESA originates from the promotion of ground-state electrons to localized excited states. Ground-state bleaching and stimulated emission signals were not evident, most likely masked by overlapping ESA contributions. For PIS, ESA bands centered at 590, 630, and 690 nm were visible at 10 ps, but disappeared after 1000 ps. The signal peaks at 590 and 630 nm primarily reflect the formation of delocalized holes. Despite the reduction in peak intensity, the decay behavior becomes monotonic and

relatively slow, reflecting a lower population of intermediate excited states and extended carrier lifetimes in PIS compared with the PI counterpart.

The ultrafast dynamics at 790 nm were further evaluated using a triple-exponential decay model[52]. With prolonged probe time, the intensity of fs-TA signals of all samples significantly weakened, indicating that the active photogenerated charge decreased. Generally, $\tau_1$ and $\tau_2$ were assigned to the trapping rates of carriers by trap states, and $\tau_3$ reflected the electron interlayer migration (Fig. 3c, f)[53]. For PIS, $\tau_1$ is 13.64 ps (54.40%), $\tau_2$ is 26.47 ps (30.60%), and $\tau_3$ is 457.16 ps (15.01%). For PI, $\tau_1$ is 16.95 ps (46.00%), $\tau_2$ is 69.91 ps (33.67%), and $\tau_3$ is 226.26 ps (20.33%). Notably, the high polarity results in an increased migration rate of charge carriers within the D-A structure. Meanwhile, the polar channel reduces the inter-layer charge recombination of 85% of the charges. Thus, the polar ketone and sulfonyl groups in this system can enhance the charge polarization, improving charge separation efficiency in COFs[54].

Due to the improved charged transport pathway, the transport of charge carriers will be enhanced. As shown in Fig. 3g and Supplementary Table 3, the effective masses of holes and electrons were estimated to be $1.422 \times 10^{-8}$ and $3.222 \times 10^{-9}$, which were lower than

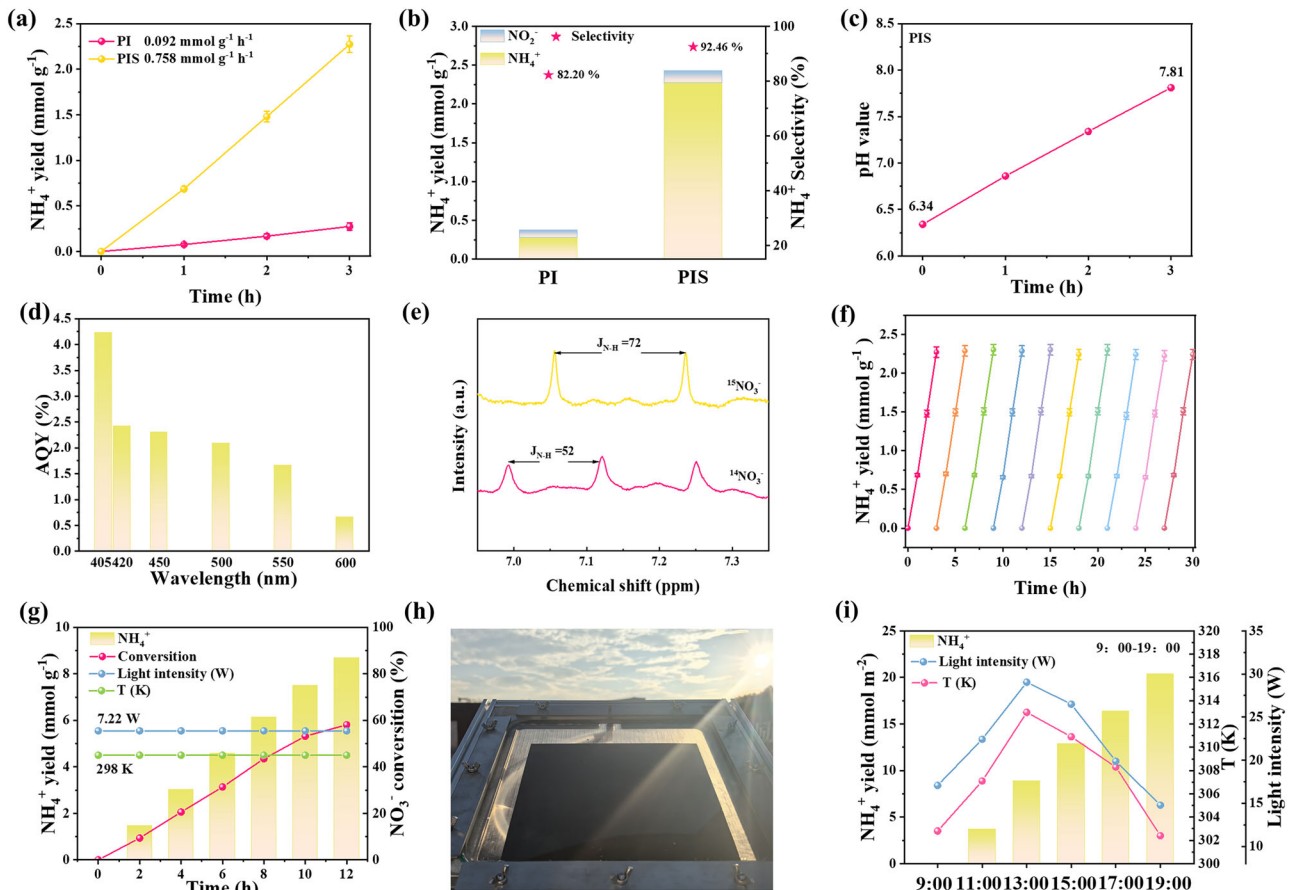

**Fig. 4 | The performance of photocatalytic nitrate reduction to ammonia.**
**a** Photocatalytic performance of $NH_4^+$ production of as-prepared samples (three measurements). **b** Selectivity of $NO_3^-$ reduction products for as-prepared samples. **c** The pH variation of the solution during the photocatalytic nitrate reduction process. **d** AQY of $NH_4^+$ production over PIS (photocatalytic: 10 mg; duration of light: 3 h; single measurement). **e** $^1H$ NMR spectra of $NO_3^-$ reduction reaction solution (using $^{14}NO_3^-$ and $^{15}NO_3^-$ as N source respectively). **f** Cyclic performance testing compared the $NH_4^+$ yield of PIS under the light condition. **g** Performance of photocatalytic $NH_4^+$ production on PIS for 12 h (temperature: 298 K, Light intensity: 7.22 W). **h** Top-down view of a 625 cm$^2$ solar panel reactor designed for $NH_4^+$ production. **i** The performance of photocatalytic production of $NH_4^+$ on PIS under sunlight over a 10-hour period.

those of PI ($7.833 \times 10^{-7}$, $7.054 \times 10^{-8}$). For PI, the carrier masses of holes and electrons are 55.08 and 21.89 times higher than for PIS. As illustrated in Supplementary Fig. 21, the reduced fluorescence intensity of PIS suggests a decreased charge recombination rate, contributing to a higher degree of electron-hole participation in photocatalytic redox processes. Both steady-state and time-resolved photoluminescence (PL) techniques were applied to study the excitation behavior, as shown in Fig. 3h and Supplementary Table 4. The fitted fluorescence lifetime of PIS is 4.877 ns, which is longer than that of PI (4.485 ns), indicating an improvement in singlet exciton dissociation. Figure 3i and Supplementary Fig. 22 depict the normalized intensity of PL emission as a function of temperature for PI and PIS, respectively. For PIS, the singlet exciton binding energy is estimated to be 124.58 meV, significantly lower than the 201.68 meV observed for PI. Such reduction facilitates exciton dissociation into free carriers[55,56]. Electrochemical impedance spectroscopy (EIS) was additionally employed to assess charge transfer properties, with the semicircle diameter in the Nyquist plot corresponding to charge-transfer resistance. Supplementary Fig. 23 indicates that PIS has a much smaller curvature radius than PI, enabling charge transfer with lower impedance. Meanwhile, the photocurrent response, reflecting photoelectric-conversion efficiency, shows that PIS exhibits a significantly enhanced photocurrent response compared to PI (Supplementary Fig. 24), suggesting more free charges are available on the PIS surface for reactions.

## Photocatalytic nitrate reduction

The photocatalytic reduction activity of the prepared samples for converting $NO_3^-$ to ammonia was also evaluated in a circulating condensate water system. Notably, the concentration of nitrate ion is 0.99 mM, which falls within the low-concentration range for urban wastewater treatment[9,57]. In this system, we also evaluated the activity of the prepared samples by photocatalytic reduction of $NO_3^-$ to via Nessler's, ion chromatography and gas chromatography (Supplementary Figs. 25−27). As depicted in Fig. 4a, the photocatalytic PI and PIS exhibited average $NH_4^+$ generation rates of 0.092 ($\pm 0.013$) and 0.758 ($\pm 0.03$) mmol g$^{-1}$ h$^{-1}$, respectively. Additionally, the $NH_4^+$ selectivity were 82.20% for PI and 92.46% for PIS (Fig. 4b). These results are in line with the pH values presented in Fig. 4c. Specifically, although they can produce nitrite ions, the final $NO_2^-$ concentration was below the detection limit of 0.3 mg L$^{-1}$, which is well below the discharge threshold (1.0 mg L$^{-1}$) specified in typical wastewater standards. It is worth noting that PIS maintained the highest $NH_4^+$ production rate, 8.24 times higher than that of PI, demonstrating its highly competitive performance. The apparent quantum yield (AQY) of PIS was calculated to be 4.23% at 405 nm, indicating high photon utilization (Fig. 4d). Furthermore, to investigate the nitrogen source, we conducted isotope experiments (Fig. 4e and Supplementary Fig. 28). It was verified that the nitrogen atom incorporated into the main product, $NH_4^+$, came directly from $NO_3^-$. Moreover, after 10 reaction cycles, the

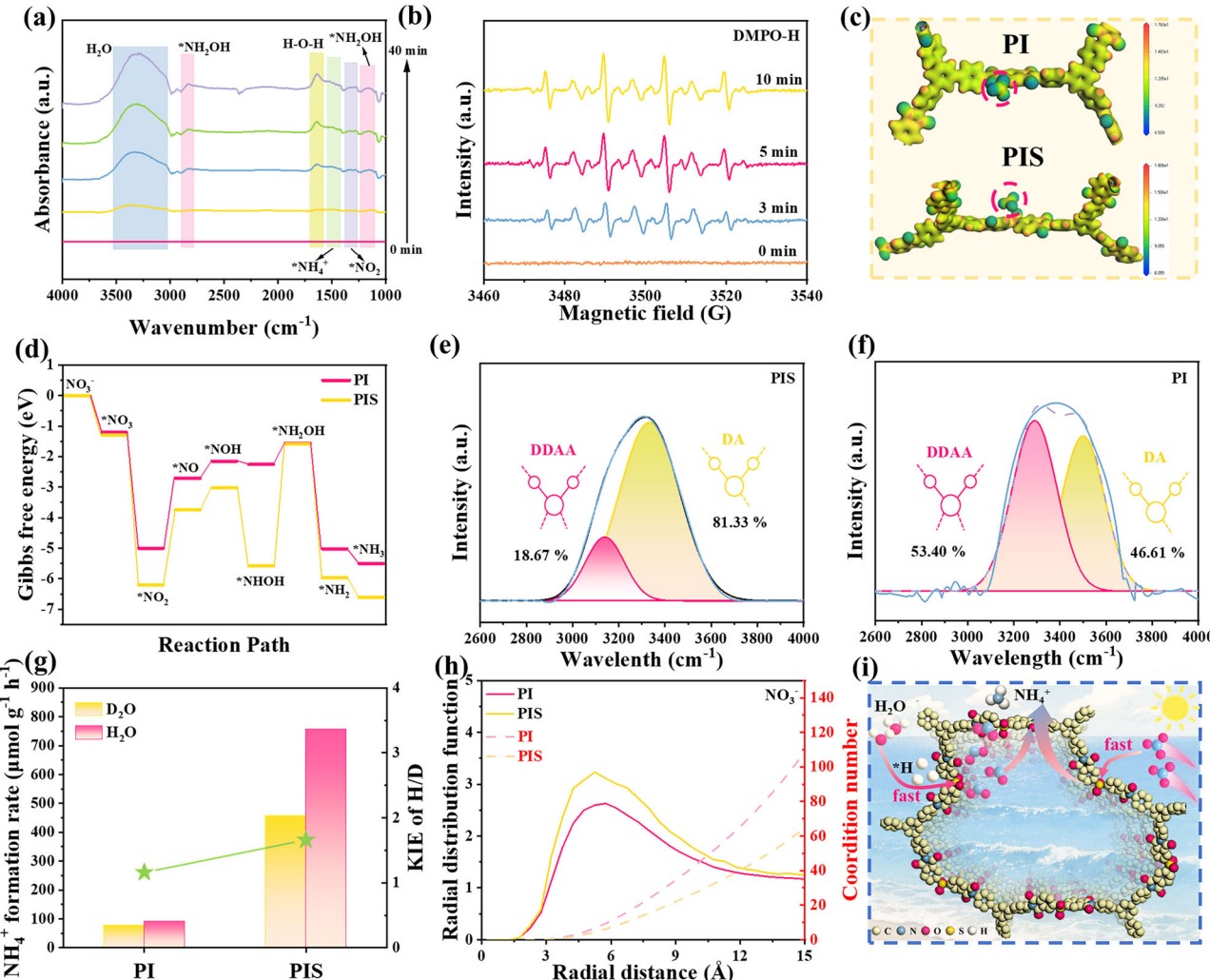

**Fig. 5 | Research on the mechanism of photocatalytic nitrate reduction. a** In-situ FTIR spectrum obtained from PIS under $NO_3^-$ solution conditions at different times. **b** Operando EPR spectra of solutions collected after 10 min of photocatalytic treatment using the PIS system in 1.0 M KOH electrolyte. **c** Comparison of the absolute total charge density of $NO_3^-$ adsorbed on PI and PIS. **d** Gibbs free energy for photocatalytic $NO_3^-$ reduction progress. In-situ FTIR analysis of water adsorption patterns using the $H_2O$-$NO_3$ system adsorbed on (**e**) PIS and (**f**) PI. **g** KIE of H/D over PI and PIS. **h** Radial distribution functions and coordination numbers of PIS and PI in $NO_3^-$. **i** Mechanism diagram of nitrate reduction to ammonia over PIS.

photocatalytic activity and structure of the PIS system showed almost no decay (Fig. 4f and Supplementary Figs. 29–31), fully demonstrating its high structural stability during the photocatalytic reduction of $NO_3^-$ to ammonia. In order to evaluate the true active performance of the material, we conducted tests under sunlight conditions. Firstly, we conducted a simulation in the experimental environment of 12 h (Fig. 4g). Notably, the average $NH_4^+$ generation rate of PIS can be calculated to be 8.694 mmol $g^{-1}$, and when expressed as nitrate nitrogen ($NO_3^-$-N), the residual $NO_3^-$ concentration is 9.44 mg $L^{-1}$, meeting the discharge standard requirements. The immobilization of particulate photocatalysts represents a critical challenge for large-scale $NO_3^-$ reduction. The large-scale carbon paper was installed in a proof-of-concept reactor consisting of 0.25 m × 0.25 m tanks. As shown in Fig. 4h, i, the cumulative $NH_4^+$ concentration increased linearly over 10 hours, and the formation rate changed in accordance with the variations in temperature and light. Finally, the $NH_4^+$ generation of PIS can reach 20.363 mmol $m^{-2}$, and the residual $NO_3^-$ concentration is 5.98 mg $L^{-1}$, meeting the discharge standard requirements[58]. More importantly, the above results fully demonstrate the broad and practical efficacy of the photocatalyst PIS in environmental pollutant degradation and energy conversion.

## Exploration of the photocatalytic mechanism

To clarify the reaction mechanism, PIS and PI were examined via in-situ FTIR during nitrate reduction, with the 3301 $cm^{-1}$ peak (Fig. 5a and Supplementary Fig. 32) corresponding to $H_2O$. The bands at 2833 $cm^{-1}$ and 1160 $cm^{-1}$ were related to *$NH_2OH$ adsorption[59,60]. The intermediate products $NH_4^+$ and *$NO_2$ were observed at 1476 and 1306 $cm^{-1}$, respectively, indicating a decrease in the successive production and utilization of by-products throughout the $NO_3^-$ process[61,62]. The H-O-H stretching vibrations of $H_2O$ (1636 $cm^{-1}$) were markedly enhanced, verifying the effective dissociation of $H_2O$ on PIS, yielding a large amount of stabilized *H[62–64]. As depicted in Supplementary Fig. 33, the [1]H NMR spectrum of PIS revealed the presence of hydroxylamine intermediates, while the [15]N NMR spectra provided evidence for the existence of nitrite intermediates. This finding confirmed that hydroxylamine was the crucial intermediate in the reaction product and that the nitrogen in the reaction stemmed from the supplied nitrate ions[65]. Based on our nitrate ammonification hypothesis and previous studies, PIS first adsorbs $NO_3^-$ and subsequently converts it to $NH_4^+$.

To assess the adsorption performance of the synthesized catalysts, contact angle testing was carried out on each of the samples (Supplementary Fig. 34)[66]. PIS had lower contact angles in both water

and nitrate solutions than PI. This means amide substitution with a stiffer structure enhances active-site accessibility, accelerating $NO_3^-$ reduction to ammonia, and PNRA involves deoxygenation followed by hydrogenation. It's evident that active hydrogen holds a pivotal position in both steps. In our system, Cyclic Voltammetry (CV) measurements (Fig. 5b and Supplementary Figs. 35 and 36) and Electron Paramagnetic Resonance (EPR) measurements with and without the presence of $NO_3^-$ validated the formation and adsorption of *H on PIS[67–69]. To find out how *H works in the PNRA process, we did *H quenching tests after adding tertiary butyl alcohol (TBA). As seen in Supplementary Fig. 37, the $NH_4^+$ production of PIS decreased significantly, whereas that of PI exhibited only a slight decline after adding TBA, confirming that the differentiated bipolar active sites facilitate the generation of *H.

In order to gain a clearer understanding of these reaction processes, DFT calculations were carried out. As shown in Supplementary Figs. 38 and 39, the Gibbs free energies for the adsorption of *$NO_3$ and *H on the O = S = O bond of PIS are −5.10 eV and −1.37 eV, respectively, which are lower than those for adsorption on the C = O bond (−4.81 eV and −0.84 eV). This indicates that both *$NO_3$ and *H are more readily adsorbed on the O = S = O bond of PIS, thereby facilitating the progress of the reaction. In addition, the N-O(3) average bond length and N-O(3) charge difference exhibited by PIS are larger than those exhibited by PI in the adsorption of $NO_3^-$ (Supplementary Fig. 40 and Fig. 5c), suggesting the enhanced polarity endows PIS with improved performance in adsorption and catalytic reactions. Moreover, the longer N-O bond exhibits a larger charge difference, indicating that this bond is preferentially activated and cleaved on the PIS surface, which is consistent with its enhanced polarity and catalytic activity (Supplementary Fig. 41). It is well known that deoxygenation and hydrogenation reactions play a key role in nitrate reduction (Supplementary Figs. 42 and 43). As depicted in Fig. 5d, the protonation of *$NO_2$ to form *NO requires an energy input, indicating that this step is kinetically demanding and may contribute significantly to the overall activation barrier during nitrate ($NO_3^-$) reduction. Moreover, the hydrogenation of *$NH_2OH$ is −3.45 eV and −4.38 eV, indicating that active-site design guides intermediates, enhancing selectivity and reducing activation barriers.

At ultra-low concentrations, nitric acid molecules ($HNO_3$) are highly dispersed within the solution, with minimal direct interactions among them. In this situation, the hydrogen-bonding network is mainly formed by water molecules. Nitric acid molecules connect to water via single or a few hydrogen bonds, forming an "isolated molecule-hydrogen bond bridge" structure. This configuration requires that nitric acid molecules depend on water molecule chains for proton or $NO_3^-$ transfer, making the reaction rate significantly reliant on the continuity of the hydrogen-bonding network[70,71]. In-situ FTIR of the O-H stretching region (2800-3600 $cm^{-1}$) revealed distinct hydrogen-bonding environments of interfacial water on PIS and PI, allowing identification of their coordination configurations such as bulk-like tetrahedral (DDAA) and linear (DA) structures[72]. In the DDAA configuration, each water molecule simultaneously acts as both a hydrogen bond donor and acceptor, thereby constructing a robust three-dimensional hydrogen-bonding network. Conversely, the DA structure exhibits a linear geometry with significantly weaker hydrogen bonding due to its limited donor-acceptor engagement. As shown in Fig. 5e, f, the hydrogen-bonded conformation of the single-chain form predominates (PIS for 81.33% and PI for 46.61%), which more readily exposes coordination sites to promote $NO_3^-$ adsorption. As the reaction progresses, the proportion of DDAA configurations decreases, while weaker hydrogen-bonding structures such as DA increase, indicating a transition from an ordered to a disordered hydrogen-bonding network. This is accompanied by weakened hydrogen-bond strength and enhanced water dissociation activity. This process accelerated the diffusion of nitric acid and the migration of proton

hydrogen. To prove the enhancement of proton transfer in the nitrate reduction reaction, we conducted $H_2O/D_2O$ exchange experiments to determine the kinetic isotope effect (KIE). As shown in Fig. 5g and Supplementary Table 5, the KIE values are 1.43 for PI and 1.16 for PIS, suggesting that the S = O = S group facilitates *H transfer at the catalytic interface, whereas PI exhibits slower kinetics for *H migration[73]. Moreover, the calculated activation free-energy differences ($\Delta G^{\#}$) of *H for PI and PIS are 0.88 kJ·$mol^{-1}$and 0.43 kJ·$mol^{-1}$. Therefore, the smaller $\Delta G^{\#}$ and KIE of PIS reflect its reduced sensitivity to proton transfer and a weakened interfacial hydrogen-bonding network, aligning with its enhanced ammonia production and enhanced $NO_3RR$ efficiency.

Molecular dynamics (MD) simulations were carried out to analyze the atomic structures of interfacial $H_2O$ on various material surfaces (Supplementary Data 1). MD simulations were carried out to analyze the atomic structures of interfacial $H_2O$ on various material surfaces. As illustrated in Fig. 5h and Supplementary Figs. 44 and 45, radial distribution function (RDF) analysis of the $H_2O$-$NO_3^-$ system, calculated between nitrate oxygen atoms and the surface active sites (O = S = O in PIS and C = O in PI), reveals that the first-shell peak of interfacial water molecules appears at 5.22 Å for PIS, which is shorter than that of PI (5.60 Å), indicating a more compact and ordered water network on the PIS surface. Furthermore, PIS exhibits a higher g(r) value, indicating stronger molecular trapping capacity. Meanwhile, at ultra-low concentrations, the diffusion coefficient of $NO_3^-$ on PIS ($1.54 \times 10^{-3}$ $cm^2 s^{-1}$) is significantly higher than that on PI ($5.02 \times 10^{-4}$ $cm^2 s^{-1}$, as shown in Supplementary Fig. 46). This phenomenon is ascribed to the strong polar characteristic of the surface, resulting in an uneven distribution of surface charges, thereby accelerating the adsorption of nitrate ions. Moreover, the strong polar environment on the surface disrupts the strong hydrogen bond interaction among water molecules, which enhances the diffusion of nitrate ions and is beneficial for the conversion of low-concentration nitrate ions. Therefore, the more pronounced structural responsiveness of PIS further indicates that its in-plane and out-of-plane polarities induce a locally confined micro-environment. Moreover, it inherently maintains redox selectivity through efficient carrier separation and polarity-induced charge confinement, ensuring that the reduced products will not be oxidized by holes. Additionally, it facilitates N-O bond activation, enhances the proton transfer rate, and accelerates the sequential reduction pathway of nitric acid (Fig. 5i).

## Discussion

In brief, we have proposed a straightforward approach for preparing donor units in TAPB-based covalent organic frameworks (COFs), utilizing pyromellitic dianhydride and 5,5'-sulfonylbis(isobenzofuran-1,3-dione) as linking units. This study proposes and validates an unconventional asymmetric spatial polarity strategy that precisely regulates charge-migration pathways at the molecular scale by introducing structurally differentiated polar groups into D-A COFs. Unlike traditional multicomponent strategies that result in stacking disorder and carrier dissipation, this approach constructs stable and efficient polar channels by enhancing in-plane polarity, weakening interlayer coupling, and inducing longitudinal polarization. Consequently, it enables the cooperative and directional migration of charges both within and between layers. Meanwhile, the highly polar surface not only facilitates the cleavage of N-O bonds and the reconstruction of the water hydrogen-bonding network but also significantly enhances the adsorption and diffusion capabilities for trace nitrate ions. Consequently, PIS exhibits highly active photocatalytic nitrate reduction activity, achieving an $NH_4^+$ production rate of 0.758 mmol $g^{-1}$ $h^{-1}$ without the assistance of cocatalysts or sacrificial compounds. Although DFT calculations offer valuable mechanistic insights, the simplified models and idealized conditions employed may not adequately represent the complexity of working catalytic systems. Overall, this work not only underscores the pivotal role of polarity engineering in

modulating charge behavior and interfacial reactions of photo-catalysts but also proposes an innovative design strategy for efficient nitrate reduction under trace-concentration conditions.

## Methods

### Synthesis of TAPB

In a 500 mL round-bottomed flask, 1,3,5-tribromobenzene (6.37 mmol), 4-aminophenylboronic acid pinacol ester (25.48 mmol), $K_2CO_3$ (254.96 mmol) and $Pd(PPh_3)_4$ (0.637 mmol) are added to the mixture of tetrahydrofuran (150 mL) and $H_2O$ (50 mL) is combined with the reagents, refluxed at 100°C under $N_2$ for 24 h, then cooled, extracted with ethyl acetate, washed with brine, dried over $Na_2SO_4$, and concentrated to yield 1,3,5-tris-(4-aminophenyl)benzene (TAPB).

### Synthesis of PI

TAPB (2.0 mmol), homophthalic dianhydride (3.0 mmol), anhydrous zinc acetate (2.0 mmol), and imidazole (5.0 g) were dissolved in a 500 mL three-necked flask. The mixture is heated and stirred at 140 °C for 5 h. After cooling the reaction mixture to room temperature, 250 mL of 1 M HCl is added and the mixture is stirred until no further precipitation is detected. The resulting product is named PI after final centrifugation and drying.

### Synthesis of PIS

TAPB (2.0 mmol), 5,5'-sulfonylbis (isobenzofuran-1,3-dione) (3.0 mmol), anhydrous zinc acetate (2.0 mmol), and imidazole (5.0 g) were dissolved in a 500 mL three-necked flask. The mixture was heated and stirred at 140 °C for 5 h. After cooling the reaction mixture to room temperature, 250 mL of 1 M HCl was added, and the mixture was stirred until no further precipitation was detected. The final product obtained by centrifugal drying was named PIS.

### Chemicals and materials

1,3,5-Tribromobenzene (Adamas life, ≥99%), $C_{12}H_{18}BNO_2$ (Adamas, 98% +), $K_2CO_3$ (Adamas, 99.99%), $Pd(PPh_3)_4$ (Adamas, ≥99.9%), THF (Adamas, 99.8%), EA (Adamas, 99.8%), $Na_2SO_4$ (Greagent, ≥ 99.5%), $C_{10}H_2O_6$ (Adamas, 98% +), $C_4H_6O_4Zn$ (Adamas, 99.99%), $C_3H_4N_2$ (Adamas, 99%), HCl (Carl Roth, 37%), $C_{16}H_6O_8S$ (Adamas, 99%), KOH (Adamas, 99.999%).

### Characterizations

Powder X-ray diffraction (XRD) patterns were collected using an X-ray diffractometer (Bruker D8 Advance) equipped with Cu Kα radiation. The chemical composition and valence states of the elements in the synthesized photocatalysts were analyzed by X-ray photoelectron spectroscopy (XPS, Thermo Fisher K-Alpha Plus) employing a mono-chromatized Mg Kα X-ray source (hν = 1283 eV). Fourier transform infrared (FT-IR) spectra were recorded on a Shimadzu spectrometer to identify the characteristic functional groups. The morphology and microstructure of the samples were examined by transmission electron microscopy (TEM, JEM-2100F) operated at 200 kV, while surface morphology was further characterized using a field-emission scanning electron microscope (FE-SEM, Hitachi S4800). Solid-state $^{13}C$ NMR spectra were obtained on a JNM-ECZ600R spectrometer. Electrochemical measurements were carried out using a CHI660E electrochemical workstation (Chenhua) in a conventional three-electrode configuration.

### DFT calculations

All density functional theory (DFT) calculations were performed using the CASTEP and DMol3 modules in Materials Studio. To accurately describe exchange-correlation interactions, density functional theory calculations were performed within the generalized gradient approximation using the Perdew-Burke-Ernzerhof (PBE) functional. Long-range van der Waals interactions were treated using Grimme's DFT-D3

method with Becke-Johnson damping (DFT-D3(BJ)). In CASTEP, a plane-wave basis set with a kinetic energy cutoff of 400 eV was applied. Ultrasoft pseudopotentials were used to describe the interaction between core and valence electrons. The Brillouin zone of all samples was sampled with a $2 \times 2 \times 1$ k-point grid. Geometry optimizations were carried out until the total energy, maximum force, and atomic displacement were smaller than $1.0 \times 10^{-5}$ eV/atom, 0.03 eV/Å, and 0.001 Å, respectively. A vacuum thickness and residual atomic forces were set as 20.00 Å and 0.05 eV/Å. For DMol3 calculations, a double Numerical plus d-functions (DND) basis set was adopted with a real-space cutoff radius of 3.5 Å and the Brillouin zone of all samples was set as $1 \times 1 \times 1$ k-point grids, and atomic displacements were smaller than $1.0 \times 10^{-5}$ Ha, 0.002 Ha/Å, and 0.005 Å, respectively. Regarding the computational models, both molecular and periodic models were used depending on the purpose: ① The single-layer molecular models of PI and PIS were employed to analyze frontier orbitals, electrostatic potential maps, and dipole moments. ② The periodic models were used for geometry optimization and electronic structure calculations under periodic boundary conditions. ③ To explore interlayer charge transport, bilayer models were constructed to evaluate charge density differences, dipole moments, and polarization characteristics. The same PBE functional was used for exchange-correlation. The SCF tolerance was set to $1.0 \times 10^{-6}$ Ha. Specifically, the optimized PIS structure exhibits a hexagonal unit cell with parameters a = b = 45 Å, c = 26 Å, α = β = 90°, γ = 120°, while the PI structure shows a = b = 36 Å, c = 24 Å, α = β = 90°, γ = 120°. The partial charge density and electrostatic potential distributions were obtained from Mulliken population analysis.

### Free energy calculations

The Gibbs free energy change (ΔG) associated with each elementary step in the nitrate-to-ammonia reduction pathway was evaluated using the standard thermodynamic relation:

$$\Delta G = \Delta E_{DFT} + \Delta E_{ZPE} - T\Delta S$$

where $\Delta E_{DFT}$ is the electronic energy difference obtained from DFT total energy calculations, $\Delta E_{ZPE}$ is the zero-point energy correction, and TΔS represents the entropy contribution at 298.15 K.

### MD simulations

Molecular dynamics (MD) simulations were performed using the Forcite Plus module in Materials Studio. The COMPASS III force field was employed to accurately describe both the organic framework and solvent interactions. All simulations were carried out in the NVT ensemble, with the system temperature maintained at 298 K using a Nosé-Hoover thermostat. A time step of 1.0 fs was applied, and each simulation was propagated for a total duration of 10 ns. Periodic boundary conditions were imposed in all three spatial directions. Nonbonded interactions were truncated at a cutoff distance of 12 Å, while long-range electrostatic interactions were treated using the particle mesh Ewald (PME) method with an error tolerance below $10^{-5}$. Atomic configurations were recorded every 1 ps throughout the production run, yielding approximately 10000 snapshots. The final simulation cell consisted of 100 water molecules and 5 nitrate ions. Radial distribution functions (RDFs) and related statistical analyses were calculated based on the equilibrated segment of the trajectory, specifically the final 3 ns of the simulation.

### Photocatalytic reaction

Photocatalytic nitrate reduction experiments were conducted by dispersing 10 mg of catalyst into 100 mL of an aqueous $KNO_3$ solution (100 mg L$^{-1}$). Prior to illumination, the suspension was magnetically stirred in the dark for 30 min to establish adsorption-desorption equilibrium. A 300 W xenon lamp equipped with a cutoff filter

($\lambda \geq 420$ nm, model XE300WF, irradiance 0.2 W cm$^{-2}$) was used as the visible-light source. During the reaction, 5 mL aliquots were withdrawn at 1 h intervals for analysis. The concentration of $NH_4^+$ was quantified using the Nessler's reagent method, while $NO_3^-$ and $NO_2^-$ concentrations were determined by ion chromatography (ICS-900). The formation of ammonia was further confirmed by $^1$H NMR spectroscopy.

The selectivity and conversion rate are calculated using the following equation:

$$\text{Conversion}[\%] = [(NO_3^-)_0 - (NO_3^-)_t]/(NO_3^-)_0 \times 100\%$$
$$\text{Selectivity}(\%) = NH_3/[(NO_3^-)_0 - (NO_3^-)_t] \times 100\%$$

### Detection Methods of $^{14}NH_4^+$ and $^{15}NH_4^+$

For isotopic labeling experiments, 10 mg of catalyst was added to 100 mL of an aqueous solution containing either $^{14}NO_3^-$ or $^{15}NO_3^-$ at a concentration of 0.1 g L$^{-1}$. The reaction was carried out under continuous stirring and light irradiation for 3 h. After completion, the catalyst was separated by centrifugation. Subsequently, 100 μL of the supernatant was transferred into a 1 mL centrifuge tube, followed by the addition of 500 μL of DMSO-d6 and a small amount of sulfuric acid. The mixture was sonicated to ensure uniform mixing and then transferred to an NMR tube for $^1$H NMR analysis.

### Electrochemical analysis

All electrochemical measurements were conducted in a single-cell configuration (not a membrane-separated H-type cell) using a standard three-electrode setup on a CHI660E electrochemical workstation (Chenhua). Photoelectrochemical tests were carried out using an electrochemical workstation. The electrolyte employed was a 0.5 M solution of anhydrous sodium sulfate (pH=7.1 ± 0.2), and illumination was provided by a 300 W xenon lamp. Fluorine-doped tin oxide (FTO) glass was utilized as the conductive substrate for the working electrode. A silver/silver chloride (Ag/AgCl) electrode served as the reference electrode, while a platinum wire was used as the counter electrode. To prepare the working electrode, a homogeneous slurry—comprising 20 mg of photocatalyst dispersed in 1 mL of ethanol along with 20 μL of naphthol—was spin-coated onto the FTO substrate. Subsequently, the coated substrate was dried at 60 °C overnight. The electrochemical properties of the material were thoroughly evaluated by measuring photocurrent response, electrochemical impedance spectroscopy, and Mott-Schottky analysis. The cyclic voltammetry (CV) measurements were performed at a scan rate of 50 mV s$^{-1}$, with 6 segments (cycles) for each test to ensure reproducibility. The electrochemical impedance spectroscopy (EIS) data were collected in the frequency range of 0.01 Hz to $1 \times 10^5$ Hz with an amplitude of 1 mV. EIS measurements were performed at an initial potential of 0.01 V with an AC perturbation amplitude of 5 mV. The frequency range was scanned from 100 kHz to 0.01 Hz, with one cycle collected in each frequency decade (0.1–1 Hz, 0.01–0.1 Hz, and 0.001–0.01 Hz). The solution resistance was extracted from the high-frequency interception of the Nyquist plot.

### Measurement of apparent quantum yield (AQY) efficiency

The AQY was measured and calculated on the basis of the following equation:

$$AQY = \frac{N_e}{N_P} * 100\% = \frac{10^9(v*N_A*K)*(h*c)}{(I*S*\lambda)} * 100\%$$

$$AQY_{405} = \frac{10^9(4.02 \times 10^{-10} \times 6.02 \times 10^{23} \times 8) \times (6.62 \times 10^{-34} \times 3 \times 10^8)}{(1.24 \times 10^{-3} \times 18.1 \times 405)} \times 100\% = 4.23\%$$

$N_e$: total electrons transferred by reaction
$N_P$: incident photon number
v: reaction rate (mol s$^{-1}$)
$N_A$: Avogadro's number (6.02 × 10$^{23}$ mol$^{-1}$)

K: number of electrons transferred by the reaction
h: Planck's constant (6.62 × 10$^{-34}$ J s)
c: the speed of light (3.0 × 10$^8$ m s$^{-1}$)
I: optical power density (W m$^{-2}$)
S: incident light area (m$^2$)
λ: Incident light wavelength (nm)

## Data availability
Data supporting the findings of this study are available within the article and supplementary information. Source data are provided with this paper.

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

## Acknowledgements

This project was financially supported by the National Natural Science Foundation of China (No. 22268015). Guizhou Science and Technology Platform foundation (No. ZSYS [2025]–033). The authors would like to thank Scientific Compass (www.shiyanjia.com) for materials characterizations and the computing support of the State Key Laboratory of Public Big Data, Guizhou University.

## Author contributions

Y.S. (Data curation, Formal analysis, Investigation, Writing –original draft), Z.W. (Formal analysis), X.D. (Data curation, Software, Writing – original draft), S.-F.Y. (Data curation, Project administration, Supervision), P.C. (Conceptualization, Formal analysis, Funding acquisition, Supervision, Writing – review & editing).

## Competing interests

The authors declare no competing interests.
