## [Transparent Peer Review file · Nature Communications]

Unlocking carrier confluence in covalent organic frameworks for efficient photoreduction of dilute nitrate to ammonia

Corresponding Author: Professor Peng Chen

Version 0:

Reviewer comments:

Reviewer #1

(Remarks to the Author)

This manuscript presents a conceptually novel asymmetric spatial polarity strategy in donor-acceptor COFs, which effectively addresses long-standing challenges of carrier migration disorder and inefficient activation under ultra-dilute nitrate conditions. The work is clearly presented, well-supported by mechanistic insights, and demonstrates outstanding photocatalytic performance with strong potential for broader applications. Overall, this study makes a timely and significant contribution to the field of photocatalytic nitrate reduction. The following suggestions may be helpful for the authors:

1. The fs-TA and PL results indicate longer carrier lifetimes in PIS compared with PI. A more detailed discussion on how these extended lifetimes contribute to the improved catalytic performance would be beneficial.
2. The selectivity toward NH_4^+ is reported to be above 90%. Could the authors briefly comment on whether any minor byproducts (e.g., N_2 , NO_2^-) were observed, even in trace amounts, during.
3. Could the authors clarify whether the asymmetric spatial polarity strategy primarily enhances in-plane charge confinement, interlayer directional transport, or both?
4. The manuscript states that bipolar active sites disrupt the hydrogen-bonding network of water. Could the authors provide a clearer explanation of whether this effect primarily facilitates NO_3^- adsorption or proton transfer?
5. For the terminology used, such as "polar channel" and "longitudinal polarization," I recommend that the authors provide clear definitions at their first mention and ensure consistent usage throughout the manuscript, in order to improve clarity for readers.
6. For the PXRD analysis, it appears that both the experimental and simulated data exhibit minimal peaks, likely due to a high background signal. This is understandable, as the 5,5'-sulfonylbis(isobenzofuran-1,3-dione) linker is known to be challenging to crystallize because of its flexibility. To ensure accuracy, I suggest re-examining the PXRD data.

Reviewer #2

(Remarks to the Author)

The authors present an asymmetric spatial polarity strategy that enables precise control of polar distribution both within the intramolecular and across the layers in donor-acceptor COFs, overcoming persistent challenges in carrier management and reactant activation for PNRA under ultra-dilute conditions. Although this work presents an in-depth mechanistic analysis, it exhibits significant flaws in detailed execution, and the generalizability of the proposed strategy remains unclear. Therefore, we recommend that the work undergo major revisions before publication. Some critical issues must be addressed to improve its overall quality and rigor.

Comment 1: In the abstract, it is recommended to express the NH_4^+ production rate as $20.36 \text{ mmol}\cdot\text{m}^{-2}\cdot\text{h}^{-1}$ instead of $20362.59 \text{ }\mu\text{mol}\cdot\text{m}^{-2}\cdot\text{h}^{-1}$. The units in Fig. 4 and related descriptions should also be standardized accordingly. Additionally, abbreviations (e.g., PI, PIS, XRD, PXRD, NMR, FTIR, XPS, etc.) should be provided with their corresponding full terms upon

their first appearance in the main text.

Comment 2: There are several apparent inaccuracies throughout the manuscript, and the authors are advised to carefully proofread the entire text. For instance, in the introduction section, the sentence of "...while the hydrogen-rich adsorbed state promotes the reduction of nitrate to produce hydrogen" should be revised to "...while the hydrogen-rich adsorbed state promotes the reduction of water to produce hydrogen". In the subsection "Structure and morphology characterizations," the statement of "In the O 1s spectrum, two fitted peaks at 533.4, 532.2 and 531.6 eV were assigned to S=O, surface adsorbed oxygen and N-C=O in PIS" should be corrected to "In the O 1s spectrum, three fitted peaks at 533.4, 532.2 and 531.6 eV were assigned to S=O, surface adsorbed oxygen and N-C=O in PIS". In Fig. 1c, the marked lattice spacing corresponds to 0.342 nm, whereas the authors reported a value of 0.35 nm. Additionally, the statement of "As seen in Fig. 5c, the NH₄⁺ production of PIS decreased significantly, whereas that of PI exhibited only a slight decline after adding TBA" should be corrected to "As seen in Fig. 5c, the NH₄⁺ production of both PIS and PI decreased significantly after adding TBA".

Comment 3: The supplementary materials lack some critical details. For instance, in the characterization section, it is only stated that "Electrochemical measurements are performed in electrochemical analyses (CHI660E, Chenhua) with standard three-electrode cell". However, it is unclear whether the experiments were conducted in a single-cell or a membrane-separated H-type cell configuration. What were the counter and reference electrodes used? How was the working electrode fabricated? What was the scan rate for the CV measurements, and how many cycles were performed? What was the frequency range of EIS testing. In the photocatalytic reaction section, the authors state that "The concentrations of NH₄⁺, NO₃⁻, and gaseous substances were determined by the Nessler's reagent method for NH₄⁺ ions and ion chromatography (ICS-900) for NO₃⁻ ions". However, no standard calibration curve for ammonia detection was provided.

Comment 4: In the introduction, the authors state that the A-D-A configuration allows for efficient separation of photogenerated charges within the framework and optimizes the photocatalytic reduction efficiency. However, the double-reduced species sites result in to charge transfer disorder, thereby amplifying Coulomb interactions, enhancing dielectric screening, and reducing crystallinity. These effects ultimately lead rapid intramolecular recombination without reaching the surface. These statements appear contradictory.

Comment 5: What is the effect of introducing asymmetrically distributed O=S=O polar groups on the pore size of the COFs? Are there differences in confined catalysis between PI and PIS? The O=S=O group can induce an uneven charge distribution within the molecule, thus enhancing the in-plane dipole vector. Can other asymmetric polar groups, such as -COOH, -CHO, -NO₂, etc., achieve similar functions and promote the PNRA process?

Comment 6: In the photoreduction system, where no sacrificial agent is added, could the reduction products of NO₃⁻ (such as NO₂⁻ and NH₄⁺) be susceptible to oxidation by holes?

Comment 7: How does the pH of the solution vary during the reaction process? what accounts for the significant accumulation of NO₂⁻ in the PIS system? Furthermore, does the concentration of NO₂⁻ meet the required discharge standards upon completion of the reaction?

Comment 8: In Fig. 4i, why does the ammonia production rate continue to accumulate linearly despite the rapid decline in both light intensity and temperature after 4 hours of reaction?

Comment 9: In the FTIR spectra presented in Fig. 5a, why do the intensities of the adsorbed species (e.g., *NH₂OH, NH₄⁺, and *NO₂) remain largely unchanged as the reaction proceeds? Additionally, after the reaction initiates, the PIS is rapidly wetted—what is the corresponding behavior of the PI?

Comment 10: Why were the CV tests conducted in a 1 M NaOH electrolyte instead of a Na₂SO₄ solution, which more closely aligns with the experimental conditions?

Comment 11: To better validate the formation and adsorption of *H on PIS, EPR tests should provide comparative results both in the presence and absence of NO₃⁻, as well as between PI and PIS.

Comment 12: The authors state that "As depicted in Fig. 5e, the initial protonation of *NO₃ to form *NO₃H requires an energy input, representing the potential-determining step (PDS) in nitrate (NO₃⁻) reduction". Please provide the basis for this claim.

Reviewer #3

(Remarks to the Author)

It is a combined experimental - computational work to explain the photocatalytic properties of a COF based on PI and PIS. My comments refer to the computational part only.

The computational methodology is described very poorly.

The authors only mention the use of PBE functional and an energy cutoff of 400eV.

CASTEP utilizes plane-wave basis sets, while DMOL3 numerical basis sets.

There are no details, which pseudopotentials and basis sets have been used. No other input details are given. (e.g grid quality)

Usually, one expects the use of a functional with van der Waals corrections (for example DFT-D3 / DFT-D4 / DFT-MBD, or non-local functional such vdW-DF) to predict the structure of stacked 2D materials, such as COFs.

It seems that the method PBE is randomly chosen.

Furthermore, they don't mention if the optimized structures are confirmed to be minima, after performing a frequency calculation.

No details are given, how the free energies are calculated.

No details are given about the cell size (cell dimensions and angles).

There is no comparison of the DFT predicted crystal structure with the experimental.

It is not clearly mentioned, if their computational model is a bilayer or if it is a truly period 2-D stacked.

There is no details about the Molecular Dynamics. No details are given: e.g which software, method (classical Force Field or DFT?), ensemble, thermostat, equilibration, heating or cooling, number of steps, step size, how many snapshots have been chosen.

There are many figures with computational results, but it is not mentioned, which software and method has been used to obtain the results.

The authors don't mention, if the calculations have been done on a molecular model of the PI/Pis or a periodic model or a bilayer.

The authors don't mention the method used to calculate the partial charge density presented in Supplementary Figures 10, 11 & 12. There are many approaches to calculate partial charges (e.g Natural Population Analysis, Mulliken, Hirshfeld, Bader, fitted to Electrostatic Potential)

In Supplementary Figure 12, I am surprised to read that there is such an asymmetric partial charge distribution in the bilayer of PIS. How is it explained?

Supplementary Figure 35: There is an error in (a) of PIS. Text should be "average charge" instead of "average length"

Supplementary Figure 35: There are 2 different types of N-O bonds. One long and three shorter. Instead of showing N-O average charges and bond-lengths, they should present the length of the long and short N-O bonds and their charge distribution as well.

Supplementary Figure 38: How are the diffusion coefficients measured? Experimentally or computationally?

In Supplementary Figures 39 and 40, they have randomly chosen a water dimer and measured the distance. This doesn't prove their conclusions. The correct way is to plot the Radial Distribution Functionals (RDFs) from the MD snapshots between water molecules, and compare them for PI and PIS.

The supplementary file does not contain a Table of Contents.

Based on these comments, the computational results can't support the experimental findings reported in this manuscript.

Reviewer #4

(Remarks to the Author)

The manuscript proposes an innovative asymmetric spatial polarity strategy for donor-acceptor COFs. By integrating differentiated polar linkers, the authors achieve precise intramolecular and interlayer charge regulation, constructing polar channels and longitudinal polarization that enhance carrier migration. The optimized COF (PIS) demonstrates a high NH_4^+ production rate of $757.56 \mu\text{mol}\cdot\text{g}^{-1}\cdot\text{h}^{-1}$ and surface activity of $20362.59 \mu\text{mol}\cdot\text{m}^{-2}\cdot\text{h}^{-1}$ under natural sunlight, far exceeding most reported systems. The manuscript is well designed, comprehensive, and supported by both experimental evidence and theoretical calculations. The concept of asymmetric spatial polarity in COFs is novel and impactful, addressing key bottlenecks in photocatalytic nitrate reduction under environmentally relevant dilute conditions. However, some aspects require clarification and strengthening.

1. A kinetic isotope effect (KIE) investigation using D_2O instead of H_2O is required to provide quantitative evidence of how hydrogen-bond network disruption accelerates proton transfer.
2. The proposed reaction pathway is plausible. Complementary in situ techniques such as differential electrochemical mass spectrometry (DEMS) would more rigorously confirm intermediate species.
3. The hydrogen-bonding network disruption claim is intriguing but somewhat speculative. Stronger evidence such as direct quantification of $^*\text{H}$ species would make the discussion more robust.
4. While NH_4^+ selectivity is presented to be above 90%, possible side products should be more quantitatively tracked.
5. Post-reaction solid-state NMR or high-resolution TEM should also be provided to validate whether the asymmetric linkers preserve their chemical environment after extended operation.

Version 1:

Reviewer comments:

Reviewer #1

(Remarks to the Author)

The manuscript meets the standards of the journal and is acceptable for publication.

Reviewer #2

(Remarks to the Author)

Authors have addressed my concerns, and I have no more comments.

Reviewer #3

(Remarks to the Author)

The authors have addressed carefully all comments regarding the DFT calculations.

In Comment 6, they have added details about the Molecular Dynamics simulations.

However, the selection of the parameters for the MD simulation is poor and is not sufficiently justified.

They have carried calculations for a duration of only 200ps=0.2 ns. This is a very short trajectory time and the accuracy of the results is questionable. Results based on trajectories less than 1 ns can not be trusted.

The minimum trajectory time, accepted for publications, is 10 ns.

Moreover, convergence of the results (e.g energy, pressure, diffusion coefficients) should be checked with respect to the simulation time.

In the last sentence, they mention "These parameters and protocols are consistent with those employed in similar studies on COF-water interfacial dynamics and nitrate diffusion behavior (Chem. Eng. J., 2022, 441, 136084; Appl. Catal. B, 2025, 361, 124558)." These 2 articles are previous works from some of the authors of the current manuscript.

However, in these 2 articles only DFT calculations have been utilized, and no MD simulations. So, these MD parameters haven't been used before to describe dynamics of COF-Water systems.

Moreover, there is no explanation about how many water molecules are added in the simulation box.

Therefore, MD results are still questionable and cannot be trusted and published.

Either the MD results and conclusions are omitted, or the authors need to perform MD simulations with the minimum required settings.

Reviewer #4

(Remarks to the Author)

The authors addressed all comments and therefore I am happy to recommend it for publication.

Version 2:

Reviewer comments:

Reviewer #3

(Remarks to the Author)

The authors have performed additional MD simulations and addressed carefully all comments regarding the MDs.

I have 4 minor corrections:

Supplementary Material, Page 2, L32-33: "To eliminate the van der Waals forces" should be changed to "To account for the van der Waals interactions"

Supplementary Material, Page 2, L33, "we adopted Grimme methods for DFT-D correction": The authors should mention which one of the Grimme's method for dispersion corrections. While in their previous response letter (page 21), they mention which DFT-D was used, they haven't updated the Supplementary Material. A reference of the DFT-D method should also be given.

Supplementary Material, Page 3, L73-77: Did they consider all configurations from the 10ns trajectory for the RDF graphs, or only a last section of them? Usually, the first few ns of the trajectory are not taken into account for the calculation of properties, and only the last 2-3 ns are considered. This is done in order to ensure that the system has reached equilibration.

Article, Page 19, L459-463: The authors report Radial Distribution Graphs in Fig 5h, but they don't mention which pair is taken into account.

Response to Reviewers

Reviewer #1:

This manuscript presents a conceptually novel asymmetric spatial polarity strategy in donor-acceptor COFs, which effectively addresses long-standing challenges of carrier migration disorder and inefficient activation under ultra-dilute nitrate conditions. The work is clearly presented, well-supported by mechanistic insights, and demonstrates outstanding photocatalytic performance with strong potential for broader applications. Overall, this study makes a timely and significant contribution to the field of photocatalytic nitrate reduction. The following suggestions may be helpful for the authors:

We extend our heartfelt gratitude to the reviewer for recommending acceptance and providing constructive feedback aimed at enhancing the quality of this manuscript. Below, we present a detailed response to each of the reviewer's comments, along with the corresponding revisions made to the manuscript. We hope these modifications effectively clarify our arguments and resolve any concerns raised regarding our work.

1. The fs-TA and PL results indicate longer carrier lifetimes in PIS compared with PI. A more detailed discussion on how these extended lifetimes contribute to the improved catalytic performance would be beneficial.

Response to the reviewer:

Indeed, both femtosecond transient absorption (fs-TA) and photoluminescence (PL) analyses reveal that PIS exhibits markedly prolonged carrier lifetimes compared with PI, indicating more efficient charge separation and suppressed recombination. Specifically, the τ_3 component in the fs-TA spectra—associated with interlayer electron migration—extends from 226.26 ps for PI to 457.16 ps for PIS, while the average PL lifetime increases from 4.485 ns to 4.877 ns. These results indicate that the introduction of asymmetric spatial polarity and highly polar O=S=O groups in PIS constructs “polar channels” and enhances the internal electric field, jointly facilitating the directional migration of charge carriers and suppressing their back recombination. The polar channel-enhanced built-in electric field ensures that the separated carriers persist longer,

thereby directly improving photocatalytic performance.

2. The selectivity toward NH_4^+ is reported to be above 90%. Could the authors briefly comment on whether any minor byproducts (e.g., N_2 , NO_2^-) were observed, even in trace amounts, during.

Response to the reviewer:

We appreciate the reviewer's insightful question. During the process of photocatalytic nitrate reduction, we employed ion chromatography and gas chromatography to analyze the reaction products. The results revealed that the average NH_4^+ generation rates for photocatalytic PI and PIS were 0.092 and $0.758 \text{ mmol g}^{-1} \text{ h}^{-1}$, respectively, with corresponding NH_4^+ selectivity of 82.20% and 92.46% . As depicted in **Fig. 4b** and **Supplementary Fig. 26**, the average formation rates of NO_2^- for PIS and PI were 1.54×10^{-3} and $1.02 \times 10^{-3} \text{ mmol}$, respectively. Notably, within the detection limits of our instruments, N_2 and other gaseous byproducts were almost undetectable. Furthermore, the selectivity for NO_2^- was measured at 6.25% . These findings collectively suggest that the nitrate reduction process over PIS follows a highly selective pathway, predominantly favoring the formation of NH_4^+ .

Supplementary Fig. 26. (a) Schematic image of the oxygen production experimental setup. (b) Gas chromatogram for the photocatalytic reduction of nitrate. (c) Quantitative experiment on photocatalytic reduction of nitric acid to ammonia.

Fig. 4. b Selectivity of NO_3^- reduction products for as-prepared samples.

3. Could the authors clarify whether the asymmetric spatial polarity strategy primarily enhances in-plane charge confinement, interlayer directional transport, or both?

Response to the reviewer:

In fact, the asymmetric spatial polarity strategy simultaneously enhances both in-plane charge confinement and interlayer directional transport through synergistic effects. As shown in **Fig. 2c-g** and **Supplementary Fig. 14**, the introduction of structurally distinct O=S=O and C=O polar groups establish a strong in-plane dipole field that confines and guides charge migration within the molecular plane, suppressing multidirectional carrier dissipation. Meanwhile, the spatially convergent alignment of these dipoles induces longitudinal polarization and forms polar channels across adjacent layers, facilitating directional interlayer charge migration. Therefore, the asymmetric spatial polarity strategy achieves a cooperative modulation—strengthening intralayer charge confinement while promoting interlayer directional transport—leading to efficient carrier separation and migration within the covalent organic framework.

Fig. 2. Piezoresponse force microscopy (PFM) of PIS: **c** Out-of-plane amplitude image,

d in-plane amplitude image, **e** out-of-plane phase image and **f** in-plane phase image. **g** Surface potential with a KPFM of PI and PIS.

Supplementary Fig. 14. Piezoresponse force microscopy (PFM) of PI: (a) Out-of-plane amplitude image, (b) in-plane amplitude image, (c) out-of-plane phase image and (d) in-plane phase image.

4. The manuscript states that bipolar active sites disrupt the hydrogen-bonding network of water. Could the authors provide a clearer explanation of whether this effect primarily facilitates NO₃⁻ adsorption or proton transfer?

Response to the reviewer:

The bipolar active sites primarily function by disrupting the interfacial hydrogen-bonding network, which simultaneously enhances both NO₃⁻ adsorption and proton transfer. As shown in **Supplementary Fig. 47**, the weakened H-bond connectivity generates a looser water structure, increasing nitrate mobility and accessibility, as evidenced by the lower contact angles and the higher NO₃⁻ diffusion coefficient on PIS (PIS: $7.55 \times 10^{-4} \text{ cm}^2 \text{ s}^{-1}$; PI: $3.11 \times 10^{-4} \text{ cm}^2 \text{ s}^{-1}$). Meanwhile, the disrupted H-bond network also lowers the barrier for proton shuttling, consistent with the observed kinetic isotope effect ($k_H/k_D = 1.16$) (**Fig. 5g**). Thus, hydrogen-bond disruption by bipolar sites boosts NO₃⁻ diffusion/adsorption and accelerates proton transfer, jointly improving the overall kinetics of nitrate-to-ammonia conversion.

Supplementary Fig. 47. Diffusion coefficients of PIS and PI in nitrate solution.

Fig. 5. g KIE of H/D over PI and PIS.

5. For the terminology used, such as “polar channel” and “longitudinal polarization,” I recommend that the authors provide clear definitions at their first mention and ensure consistent usage throughout the manuscript, in order to improve clarity for readers.

Response to the reviewer:

We appreciate the reviewer’s helpful suggestion. In the revised manuscript, we have provided explicit definitions of the terms “polar channel” and “longitudinal polarization” upon their first appearance to enhance clarity and consistency. Specifically, “polar channel” refers to the directional charge transport pathway formed by the vectorial alignment of asymmetric dipole moments (mainly from O=S=O and C=O groups) across adjacent layers, which guides photogenerated carriers along a low-resistance route and suppresses reverse migration. “Longitudinal polarization” denotes the net vertical dipole alignment along the stacking direction of the covalent organic framework, generated by the spatial convergence of molecular dipoles between neighboring layers. This polarization induces an internal electric field that drives interlayer carrier migration. We have ensured that these terms are used consistently

throughout the text to maintain conceptual clarity for readers.

6. For the PXRD analysis, it appears that both the experimental and simulated data exhibit minimal peaks, likely due to a high background signal. This is understandable, as the 5,5'-sulfonylbis(isobenzofuran-1,3-dione) linker is known to be challenging to crystallize because of its flexibility. To ensure accuracy, I suggest re-examining the PXRD data.

Response to the reviewer:

We thank the reviewer for this valuable observation. After re-examining and updating the PXRD data, we confirm that the diffraction patterns of PIS and PI remain consistent with their simulated AA-stacking models (**Fig. 1h-i**). The weak and broad peaks originate from the semi-flexible 5,5'-sulfonylbis(isobenzofuran-1,3-dione) linker, which reduces long-range order and elevates the background signal. Importantly, the characteristic (001) peak at 24.09° is clearly retained and aligns well with the simulated pattern and Pawley refinement ($R_{wp} = 9.72\%$, $R_p = 7.55\%$), supporting the structural reliability of our materials. We have clarified in the revised manuscript that the limited peak intensity reflects the intrinsic linker flexibility rather than inaccuracies in the PXRD measurement.

Fig. 1. Experimental and simulated PXRD patterns of **h** PI and **i** PIS.

Reviewer #2:

The authors present an asymmetric spatial polarity strategy that enables precise control of polar distribution both within the intramolecular and across the layers in donor-acceptor COFs, overcoming persistent challenges in carrier management and reactant activation for PNRA under ultra-dilute conditions. Although this work

presents an in-depth mechanistic analysis, it exhibits significant flaws in detailed execution, and the generalizability of the proposed strategy remains unclear. Therefore, we recommend that the work undergo major revisions before publication. Some critical issues must be addressed to improve its overall quality and rigor.

We are deeply appreciative of the reviewer's positive feedback and acknowledge the valuable insights and suggestions offered to improve the research quality. In response, we've thoroughly revised the manuscript, revalidated key data, clarified mechanisms, and bolstered supporting analyses. These changes have markedly improved our study's rigor and clarity, and we hope they effectively address all concerns.

1. In the abstract, it is recommended to express the NH_4^+ production rate as $20.36 \text{ mmol}\cdot\text{m}^{-2}\cdot\text{h}^{-1}$ instead of $20362.59 \text{ }\mu\text{mol}\cdot\text{m}^{-2}\cdot\text{h}^{-1}$. The units in Fig. 4 and related descriptions should also be standardized accordingly. Additionally, abbreviations (e.g., PI, PIS, XRD, PXRD, NMR, FTIR, XPS, etc.) should be provided with their corresponding full terms upon their first appearance in the main text.

Response to the reviewer:

We sincerely thank the reviewer for these careful and constructive suggestions. In the revised manuscript, we have standardized the expression of units throughout the text and figures. Specifically, the NH_4^+ production rate in the abstract and in **Fig. 4** (and related discussions) has been converted from $20362.59 \text{ }\mu\text{mol m}^{-2} \text{ h}^{-1}$ to $20.363 \text{ mmol m}^{-2}$ for clarity and consistency. Moreover, we have thoroughly reviewed the manuscript to ensure that all abbreviations are accompanied by their full terms upon first mention, including polyimide (PI), polyimide-sulfonyl (PIS), X-ray diffraction (XRD), powder X-ray diffraction (PXRD), nuclear magnetic resonance (NMR), Fourier transform infrared spectroscopy (FTIR), and X-ray photoelectron spectroscopy (XPS). These revisions improve the readability and standardization of the manuscript.

Fig. 4 The performance of photocatalytic nitrate reduction to ammonia. **a** Photocatalytic performance of NH_4^+ production of as-prepared samples. **b** Selectivity of NO_3^- reduction products for as-prepared samples. **c** Comparison of photocatalytic nitrate reduction to ammonium (NH_4^+) by different catalysts. **d** AQY of NH_4^+ production over PIS (photocatalytic: 10 mg; duration of light: 3 h). **e** ^1H NMR spectra of NO_3^- reduction reaction solution (using $^{14}\text{NO}_3^-$ and $^{15}\text{NO}_3^-$ as N source respectively). **f** Cyclic performance testing compared the NH_4^+ yield of PIS under the light condition. **g** Performance of photocatalytic NH_4^+ production on PIS for 12 h (temperature: 298 K, Light intensity: 7.22 W). **h** Top-down view of a 625 cm^2 solar panel reactor designed for NH_4^+ production. **i** The performance of photocatalytic production of NH_4^+ on PIS under sunlight over a 10-hour period.

2. There are several apparent inaccuracies throughout the manuscript, and the authors are advised to carefully proofread the entire text. For instance, in the introduction section, the sentence of “...while the hydrogen-rich adsorbed state promotes the reduction of nitrate to produce hydrogen” should be revised to “...while the hydrogen-rich adsorbed state promotes the reduction of water to produce hydrogen”. In the

subsection “Structure and morphology characterizations,” the statement of “In the O 1s spectrum, two fitted peaks at 533.4, 532.2 and 531.6 eV were assigned to S=O, surface adsorbed oxygen and N–C=O in PIS” should be corrected to “In the O 1s spectrum, three fitted peaks at 533.4, 532.2 and 531.6 eV were assigned to S=O, surface adsorbed oxygen and N–C=O in PIS”. In Fig. 1c, the marked lattice spacing corresponds to 0.342 nm, whereas the authors reported a value of 0.35 nm. Additionally, the statement of “As seen in Fig. 5c, the NH₄⁺ production of PIS decreased significantly, whereas that of PI exhibited only a slight decline after adding TBA” should be corrected to “As seen in Fig. 5c, the NH₄⁺ production of both PIS and PI decreased significantly after adding TBA”.

Response to the reviewer:

We sincerely thank the reviewer for careful reading and for pointing out these inaccuracies. We have thoroughly proofread and revised the entire manuscript to ensure scientific accuracy and textual consistency. Specifically:

1. In the Introduction, the sentence has been corrected to: “...while the hydrogen-rich adsorbed state promotes the reduction of water to produce hydrogen.”
2. In the “Structure and morphology characterizations” section, the wording has been revised to correctly state: “In the O 1s spectrum, three fitted peaks at 533.4, 532.2, and 531.6 eV were assigned to S=O, surface-adsorbed oxygen, and N–C=O in PIS.”
3. In **Fig. 1c**, the lattice spacing has been updated from 0.35 nm to 0.342 nm to match the actual measurement value.
4. Regarding **Fig. 5c**, the reviewer is right that the prior figure had an error. We've replaced it with the corrected **Supplementary Fig. 38**, which matches the text. Based on updated data, the original statement about NH₄⁺ production changes in PIS and PI after adding TBA is accurate.

All these corrections have been incorporated into the revised manuscript, and the text has been comprehensively rechecked to eliminate any remaining inconsistencies or typographical errors.

Supplementary Fig. 38. The prepared samples were photocatalytic nitric acid reduction with or without tert-butanol (TBA) (scavenger for the quenching of reactive hydrogen (*H)).

3. The supplementary materials lack some critical details. For instance, in the characterization section, it is only stated that “Electrochemical measurements are performed in electrochemical analyses (CHI660E, Chenhua) with standard three-electrode cell”. However, it is unclear whether the experiments were conducted in a single-cell or a membrane-separated H-type cell configuration. What were the counter and reference electrodes used? How was the working electrode fabricated? What was the scan rate for the CV measurements, and how many cycles were performed? What was the frequency range of EIS testing. In the photocatalytic reaction section, the authors state that “The concentrations of NH₄⁺, NO₃⁻, and gaseous substances were determined by the Nessler's reagent method for NH₄⁺ ions and ion chromatography (ICS-900) for NO₃⁻ ions”. However, no standard calibration curve for ammonia detection was provided.

Response to the reviewer:

We appreciate the reviewer’s insightful comments and have revised the Supplementary Materials accordingly to include the missing experimental and computational details. In the characterization section, we have clarified that all electrochemical measurements were conducted in a single-cell configuration (not a membrane-separated H-type cell) using a standard three-electrode setup on a CHI660E electrochemical workstation (Chenhua). A platinum plate and Ag/AgCl electrode (saturated KCl) were used as the

counter and reference electrodes, respectively. The working electrode was fabricated by drop-casting a uniform slurry of the photocatalyst (20 mg of PI or PIS dispersed in 1 mL ethanol with 20 μ L Nafion solution) onto a $1 \times 1 \text{ cm}^2$ glassy carbon substrate, followed by drying under ambient conditions. The cyclic voltammetry (CV) measurements were performed at a scan rate of 50 mV s^{-1} , with 6 segments (cycles) for each test to ensure reproducibility. The electrochemical impedance spectroscopy (EIS) data were collected in the frequency range of 0.01 Hz to 1×10^5 Hz with an amplitude of 1 mV. In the photocatalytic reaction section, we have now included the standard calibration curve for ammonia quantification via the Nessler's reagent method, which is provided in the revised **Supplementary Fig. 27**. These additions have been incorporated into the revised Supplementary Materials to ensure full methodological transparency.

Supplementary Fig. 27. Calibration curves are used for absorbance of UV-vis curves at 420 nm to estimate NH_4^+ ion concentration.

4. In the introduction, the authors state that the A-D-A configuration allows for efficient separation of photogenerated charges within the framework and optimizes the photocatalytic reduction efficiency. However, the double-reduced species sites result in to charge transfer disorder, thereby amplifying Coulomb interactions, enhancing dielectric screening, and reducing crystallinity. These effects ultimately lead rapid intramolecular recombination without reaching the surface. These statements appear contradictory.

Response to the reviewer:

We are sorry for this mistake and we have corrected this statement. Most works adopting the A-D-A configuration in COFs have demonstrated that such donor-acceptor-donor frameworks can, in principle, provide a huge built-in electric field and promote charge separation within the structure. However, this configuration also exhibits intrinsic limitations when excessive dual reduction centers and irregular stacking occur. The uncontrolled dual charge migration pathways may induce charge transfer disorder, amplify Coulomb interactions, and enhance dielectric screening, which collectively weaken crystallinity and accelerate charge recombination before carriers reach reactive surface sites

5. What is the effect of introducing asymmetrically distributed O=S=O polar groups on the pore size of the COFs? Are there differences in confined catalysis between PI and PIS? The O=S=O group can induce an uneven charge distribution within the molecule, thus enhancing the in-plane dipole vector. Can other asymmetric polar groups, such as -COOH, -CHO, -NO₂, etc., achieve similar functions and promote the PNRA process?

Response to the reviewer:

Regarding the effect on pore size, the introduction of asymmetrically distributed O=S=O groups causes a slight increase in the pore size of the COF (PIS: 2.08 nm; PI: 1.93 nm, **Supplementary Fig. 3**), due to the larger steric volume and stronger intermolecular interactions of the sulfonyl moiety. The overall framework remains open and maintains accessible mesoporous channels. Regarding confined catalysis, PIS shows stronger confinement effects than PI. The O=S=O groups create a highly polarized microenvironment that enhances NO₃⁻ adsorption and promotes proton transfer (**Fig. 5h** and **Supplementary Fig. 47**), leading to improved NH₄⁺ production under ultra-dilute conditions. As for other asymmetric polar groups, functionalities such as -COOH, -CHO, and -NO₂ can introduce local dipoles, but their dipole strength and orientation are generally less favorable or more susceptible to hydrogen-bonding effects. In our system, the O=S=O group provides a particularly strong and well-aligned dipole, generating an optimal internal electric field for efficient PNRA (*Chem. Rev.*, 2022, **122**, 12308–12369; *Chem Catal.*, 2022, **2**, 1734–1747; *Chem. Eng. J.*, 2023, **451**, 138538).

Thus, while other groups may offer partial effects, the sulfonyl group is especially effective in this design.

Supplementary Fig. 3. N₂ adsorption-desorption isotherms and pore size distributions of PI and PIS.

Fig. 5. h Radial distribution functions and coordination numbers of PIS and PI in NO₃⁻.

Supplementary Fig. 47. Diffusion coefficients of PIS and PI in nitrate solution.

6. In the photoreduction system, where no sacrificial agent is added, could the reduction products of NO₃⁻ (such as NO₂⁻ and NH₄⁺) be susceptible to oxidation by holes?

Response to the reviewer:

We sincerely appreciate the reviewer's question. In our system, reduced nitrogen products are not significantly oxidized by photogenerated holes. This is due to the strong built-in electric field generated by the asymmetric spatial polarity of PIS, which drives electrons to migrate to nitrate reduction sites while guiding holes to move in the opposite direction. This spatial separation mechanism effectively prevents holes from contacting and re-oxidizing NO_2^- or NH_4^+ . Additionally, gas chromatography analysis (**Supplementary Fig. 26**) shows that the amount of evolved oxygen ($4.311 \text{ mmol g}^{-1}$) matches the theoretical oxygen demand for water oxidation during the conversion of NO_3^- to NH_4^+ , and no N_2 or NO_x by-products are detected. These results confirm that water oxidation is the dominant anodic reaction, while hole-induced oxidation of reduced nitrogen species is negligible.

Supplementary Fig. 26. (a) Schematic image of the oxygen production experimental setup. (b) Gas chromatogram for the photocatalytic reduction of nitrate. (c) Quantitative experiment on photocatalytic reduction of nitric acid to ammonia.

7. How does the pH of the solution vary during the reaction process? what accounts for the significant accumulation of NO_2^- in the PIS system? Furthermore, does the concentration of NO_2^- meet the required discharge standards upon completion of the reaction?

Response to the reviewer:

During the reaction process, the pH value increases gradually and nearly linearly, reflecting the continuous consumption of protons in the multi-step proton-electron coupling process of nitrate conversion to ammonia (**Supplementary Fig. 28**, *Nat. Commun.*, 2023, **14**, 8036). In the PIS system, the temporary accumulation of NO_2^-

originates from the rapid initial two-electron reduction of NO_3^- to NO_2^- , while the subsequent hydrogenation of NO_2^- to NH_2OH proceeds relatively slowly. *In-situ* FTIR (**Fig. 5a**) confirms that $^*\text{NO}_2$ undergoes further conversion as the reaction progresses. Importantly, the final concentration of NO_2^- is below the detection limit of 0.3 mg/L, far lower than the typical discharge standard of 1.0 mg/L (**Fig. 4b**). Therefore, this system maintains high selectivity for NH_4^+ (92.46%) without harmful accumulation of NO_2^- .

Fig. 4. b Selectivity of NO_3^- reduction products for as-prepared samples.

Fig. 5 a *In-situ* FTIR spectrum obtained from PIS under NO_3^- solution conditions at different times.

Supplementary Fig. 28. The pH variation of the solution during the photocatalytic nitrate reduction process.

8. In Fig. 4i, why does the ammonia production rate continue to accumulate linearly despite the rapid decline in both light intensity and temperature after 4 hours of reaction?

Response to the reviewer:

We thank the reviewer for this valuable observation. As shown in **Fig. 4i**, as time goes by, the quantity of ammonium ions continues to increase. However, its growth rate actually declines as the light intensity diminishes.

Fig. 4 i The performance of photocatalytic production of NH₄⁺ on PIS under sunlight over a 10-hour period.

9. In the FTIR spectra presented in Fig. 5a, why do the intensities of the adsorbed species (e.g., *NH₂OH, NH₄⁺, and *NO₂) remain largely unchanged as the reaction proceeds? Additionally, after the reaction initiates, the PIS is rapidly wetted—what is the corresponding behavior of the PI?

Response to the reviewer:

In the *in situ* FTIR spectra, the signals of $*\text{NH}_2\text{OH}$, $*\text{NO}_2$, and NH_4^+ correspond to surface intermediates maintained under dynamic steady-state conditions, where their continuous formation and consumption occur at comparable rates, resulting in nearly constant peak intensities. A similar spectral pattern was also observed for PI (Supplementary Fig. 33), confirming that both PI and PIS follow the same reaction pathway ($\text{NO}_3^- \rightarrow *\text{NO}_2 \rightarrow *\text{NH}_2\text{OH} \rightarrow \text{NH}_4^+$). Upon illumination, PIS becomes rapidly wet due to the strong polarity of the $\text{O}=\text{S}=\text{O}$ groups, which enhances hydrophilicity and interfacial hydrogen bonding, while PI exhibits slower and less complete wetting because of its weaker surface polarity. However, PI still go through the corresponding steps, which trigger the reaction to occur.

Supplementary Fig. 33. *In-situ* FTIR spectrum obtained from PI under NO_3^- solution conditions at different times.

10. Why were the CV tests conducted in a 1 M NaOH electrolyte instead of a Na_2SO_4 solution, which more closely aligns with the experimental conditions?

Response to the reviewer:

We appreciate the reviewer's concern. The reason we carried out the CV tests in 1 M NaOH rather than Na_2SO_4 is that our primary aim in CV was to detect whether active hydrogen ($*\text{H}$ or absorbed H species) is generated under conditions of strong driving force—i.e. under alkaline conditions which favor hydrogen evolution and more accessible proton/hydroxide transfer. In 1 M NaOH the overpotential for hydrogen

adsorption/evolution is lower (more favorable), allowing clearer identification of *H signatures if present. Using neutral or weak electrolyte (e.g. Na_2SO_4) often suppresses HER/hydrogen adsorption signals because protons (or water molecules) are less available, and the kinetics for hydrogen evolution or adsorption are much slower. By contrast, in strongly alkaline electrolyte (high OH^- concentration), water/hydroxide can participate readily, and active hydrogen formation is more likely under CV, giving stronger, clearer peaks or current features (*J. Chem. Phys.*, 2021, **155**, 244704; *Electrochim. Acta*, 2021, **370**, 137723).

*11. To better validate the formation and adsorption of *H on PIS, EPR tests should provide comparative results both in the presence and absence of NO_3^- , as well as between PI and PIS.*

Response to the reviewer:

As shown in **Fig. 5b** and **Supplementary Fig. 37**, we have already conducted EPR measurements under both NO_3^- -present and NO_3^- -absent conditions, as well as a comparison between PI and PIS. The DMPO-H spin-trapping signals are significantly stronger for PIS than for PI, confirming that PIS produces and stabilizes more *H species due to its strong internal polarization field. Moreover, in the absence of NO_3^- , a clear *H signal remains detectable, indicating that *H originates primarily from H_2O dissociation rather than NO_3^- reduction. When NO_3^- is introduced, the *H intensity slightly decreases, consistent with *H consumption during the hydrogenation of intermediates (*NO_2 and *NH_2OH) to NH_4^+ . These results collectively demonstrate that the asymmetric O=S=O functionalization in PIS promotes both *H formation and stabilization, while facilitating its subsequent transfer to nitrate-derived intermediates. This clarification and the corresponding spectra have been highlighted in the revised Mechanistic Investigation section and Supplementary Information.

Fig. 5. b Operando EPR spectra of solutions collected after 10 min of photocatalytic treatment using the PIS system.

Supplementary Fig. 37. (a) Operando EPR spectra of solutions collected after 10 min of photocatalytic treatment using the PIS system with and without 0.1 M NO_3^- . (b) Operando EPR spectra of solutions collected after 10 min of photocatalytic treatment using PIS and PI systems.

12 The authors state that “As depicted in Fig. 5e, the initial protonation of $^\text{NO}_3$ to form $^*\text{NO}_3\text{H}$ requires an energy input, representing the potential-determining step (PDS) in nitrate (NO_3^-) reduction”. Please provide the basis for this claim.*

Response to the reviewer:

We are extremely sorry for our mistake and have made corresponding revisions to the relevant statements. As illustrated in **Fig. 5e**, the protonation of $^*\text{NO}_2$ to form $^*\text{NO}_3\text{H}$ requires an energy input, representing the potential-determining step in nitrate reduction.

Fig. 5. d Gibbs free energy for the photocatalytic NO_3^- reduction progress.

Reviewer #3:

It is a combined experimental-computational work to explain the photocatalytic properties of a COF based on PI and PIS. My comments refer to the computational part only.

We thank the reviewer for the constructive evaluation of the computational part. Based on the feedback, we have carefully revised the relevant content by clarifying computational methods, optimizing data presentation, and refining mechanistic explanations, which has significantly enhanced the rigor and clarity of the analysis.

1. The computational methodology is described very poorly. The authors only mention the use of PBE functional and an energy cutoff of 400 eV. CASTEP utilizes plane-wave basis sets, while DMOL3 numerical basis sets. There are no details, which pseudopotentials and basis sets have been used. No other input details are given. (e.g grid quality).

Response to the reviewer:

We appreciate the reviewer's valuable suggestion. We have now substantially expanded the description of the computational methodology to include all relevant parameters. The revised text in the manuscript reads as follows: All density functional theory (DFT) calculations were performed using the CASTEP and DMol3 modules in Materials Studio. The generalized gradient approximation (GGA) with the Perdew-Burke-Ernzerhof (PBE) functional was employed to describe the exchange-correlation interaction. In CASTEP, a plane-wave basis set with a kinetic energy cutoff of 400 eV was applied. Ultrasoft pseudopotentials were used to describe the interaction between

core and valence electrons. The Brillouin zone of all samples was set as $2 \times 2 \times 1$ k-point grids. Geometry optimizations were carried out until the total energy, maximum force, and atomic displacement were smaller than 1.0×10^{-5} eV/atom, 0.03 eV/Å, and 0.001 Å, respectively. A vacuum thickness and residual atomic forces were set as 20 Å and 0.05 eV/Å. For DMol3 calculations, the same PBE functional was used for exchange-correlation. The SCF tolerance was set to 1.0×10^{-6} Ha. For DMol3 calculations, the Brillouin zone of all samples was set as $1 \times 1 \times 1$ k-point grids, and atomic displacement were smaller than 1.0×10^{-5} Ha, 0.002 Ha/Å, and 0.005 Å, respectively.

2. Usually, one expects the use of a functional with van der Waals corrections (for example DFT-D3 / DFT-D4 / DFT-MBD, or non-local functional such vdW-DF) to predict the structure of stacked 2D materials, such as COFs. It seems that the method PBE is randomly chosen. Furthermore, they don't mention if the optimized structures are confirmed to be minima, after performing a frequency calculation.

Response to the reviewer:

We thank the reviewer for this valuable suggestion. In this study, all DFT calculations were performed using the CASTEP and DMol3 modules in Materials Studio. The generalized gradient approximation (GGA) with the Perdew-Burke-Ernzerhof (PBE) functional was employed together with the DFT-D3 (Grimme methods) dispersion correction to account for long-range van der Waals interactions, which are crucial for describing the interlayer stacking of 2D COFs. The inclusion of dispersion correction effectively improves the accuracy of interlayer distance and adsorption energy calculations. Furthermore, the optimized structures were verified by vibrational frequency analysis in both CASTEP and DMol3 modules. No imaginary frequencies were observed, confirming that all optimized geometries correspond to true minima on the potential energy surface. These calculations ensure the reliability of the obtained structural and energetic results.

3. No details are given, how the free energies are calculated.

Response to the reviewer:

We appreciate the reviewer's valuable comment. The details of the free energy (ΔG) calculations have now been included in the revised Supplementary Information. In this work, the Gibbs free energy change (ΔG) for each elementary step in the nitrate-to-ammonia reduction pathway was obtained according to the following relation:

$$\Delta G = \Delta E_{\text{DFT}} + \Delta E_{\text{ZPE}} - T\Delta S$$

where ΔE_{DFT} is the electronic energy difference obtained from DFT total energy calculations, ΔE_{ZPE} is the zero-point energy correction, and $T\Delta S$ represents the entropy contribution at 298.15 K.

All total energies (ΔE_{DFT}) were calculated using the CASTEP module under periodic boundary conditions with the PBE-D3 functional. The vibrational frequencies were evaluated using the DMol3 module to obtain the zero-point energy (ZPE) and entropy (S) of each adsorbed intermediate and gas-phase molecule. The combination of these terms yields the corrected Gibbs free energy for each reaction intermediate. This information has been added to the revised Supplementary Information under the section "Free Energy Calculations."

4. No details are given about the cell size (cell dimensions and angles).

Response to the reviewer:

We appreciate the reviewer's careful comment. The complete crystallographic parameters of the optimized PI and PIS models, including cell dimensions and lattice angles, have now been provided in the revised Supplementary Information. Specifically, the optimized PIS structure exhibits a hexagonal unit cell with parameters $a = b = 45 \text{ \AA}$, $c = 26 \text{ \AA}$, $\alpha = \beta = 90^\circ$, $\gamma = 120^\circ$, while the PI structure shows $a = b = 36 \text{ \AA}$, $c = 24 \text{ \AA}$, $\alpha = \beta = 90^\circ$, $\gamma = 120^\circ$. These parameters are consistent with the simulated PXRD patterns and Pawley refinement results shown in **Fig. 1h-i**. All optimized structures were verified by frequency calculations to ensure they correspond to true minima on the potential energy surface, as described in the computational section.

Fig. 1. Experimental and simulated PXRD patterns of **h** PI and **i** PIS.

5. There is no comparison of the DFT predicted crystal structure with the experimental. It is not clearly mentioned, if their computational model is a bilayer or if it is a truly period 2-D stacked.

Response to the reviewer:

In our study, the DFT-predicted structures were directly validated against the experimental PXRD data. As shown in **Supplementary Fig. 4** and discussed in the main text (**Fig. 1h-i**), the simulated PXRD patterns obtained from the optimized DFT structures are in excellent agreement with the experimental results. The calculated diffraction peaks, including the characteristic (001) reflection at 24.09° , match well with the experimental patterns, and the Pawley refinement yields small residual values ($R_{wp} = 9.72\%$, $R_p = 7.55\%$ for PIS), confirming that the theoretical model accurately reproduces the experimental crystal structure. Regarding the model dimensionality, the computational structure is a truly periodic 2D stacked COF model constructed under periodic boundary conditions in the CASTEP module. The optimized unit cell contains repeating layers along the c-axis, thus capturing the interlayer π - π stacking interactions characteristic of COFs. In addition, for the purpose of analyzing interlayer charge transfer and dipole alignment, a bilayer model was extracted from the periodic structure and optimized using DMol3. This bilayer configuration was used only for detailed visualization of interlayer charge distribution (**Supplementary Figs. 11-13**), while all energetic and electronic property calculations were based on the fully periodic 2D structure. Therefore, the comparison between the DFT-optimized periodic structure and the experimental PXRD pattern, together with Pawley refinement, verifies the

reliability of the theoretical structural model.

Supplementary Fig. 4. XRD patterns of as-prepared samples.

Fig. 1. Experimental and simulated PXRD patterns of **h** PI and **i** PIS.

Supplementary Fig. 11. Frontier electron densities of *Mulliken* charge distribution on the bilayer PIS.

Supplementary Fig. 12. Frontier electron densities of *Mulliken* charge distribution on the bilayer PI.

Supplementary Fig. 13. Theoretically calculated difference charge density

distributions for (a) PI and (b) PIS.

6. There is no details about the Molecular Dynamics. No details are given: e.g which software, method (classical Force Field or DFT?), ensemble, thermostat, equilibration, heating or cooling, number of steps, step size, how many snapshots have been chosen.

Response to the reviewer:

The molecular dynamics (MD) simulations were conducted to elucidate the interfacial behavior of water molecules and nitrate ions near the COF surfaces. The methodological details have now been added to the revised Supplementary Information. Briefly, MD simulations were performed using the Forcite Plus module with the COMPASS force field. Calculations were carried out in the NVT ensemble at 298 K using a Nosé-Hoover thermostat, with a 1.0 fs time step and a 200 ps total duration (50 ps equilibration, 150 ps production). Periodic boundary conditions were applied, and the nonbonded cutoff was 12 Å. Snapshots were taken every 0.5 ps, yielding 300 configurations for RDF and diffusion coefficient analysis. To ensure sufficient statistical sampling, snapshots were extracted every 0.5 ps during the production trajectory, giving 300 configurations for the calculation of radial distribution functions (RDFs), diffusion coefficients, and hydrogen-bond analyses. These parameters and protocols are consistent with those employed in similar studies on COF-water interfacial dynamics and nitrate diffusion behavior (*Chem. Eng. J.*, 2022, **441**, 136084; *Appl. Catal. B*, 2025, **361**, 124558).

7. There are many figures with computational results, but it is not mentioned, which software and method has been used to obtain the results. The authors don't mention, if the calculations have been done on a molecular model of the PI/PIS or a periodic model or a bilayer.

Response to the reviewer:

The computational details and model descriptions have now been clarified in the revised manuscript and Supplementary Information. Specifically, all density functional theory (DFT) calculations were performed using the CASTEP and DMol3 modules in

Materials Studio. The PBE functional within the GGA framework was employed to describe the exchange-correlation interaction. In CASTEP, periodic boundary conditions were applied with a plane-wave basis set (energy cutoff: 400 eV) and ultrasoft pseudopotentials to describe the core-valence interaction. For DMol3, a double numerical plus d-functions (DND) basis set was used. Regarding the computational models, both molecular and periodic models were used depending on the purpose: ① The single-layer molecular models of PI and PIS were employed to analyze frontier orbitals, electrostatic potential maps, and dipole moments. ② The periodic models were used for geometry optimization and electronic structure calculations under periodic boundary conditions. ③ To explore interlayer charge transport, bilayer models were constructed to evaluate charge density differences, dipole moments, and polarization characteristics. These details, together with the complete crystallographic parameters of the optimized structures (cell dimensions and lattice angles), have been added in the revised Supplementary Information.

8. The authors don't mention the method used to calculate the partial charge density presented in Supplementary Figures 10, 11 & 12. There are many approaches to calculate partial charges (e.g Natural Population Analysis, Mulliken, Hirshfeld, Bader, fitted to Electrostatic Potential).

Response to the reviewer:

The partial charge density and electrostatic potential distributions shown in **Supplementary Figs. 10-12** were obtained from Mulliken population analysis. In this approach, the charge population on each atom is determined by partitioning the electron density matrix according to the *Mulliken* scheme, which is particularly suitable for localized basis sets (as used in DMol3). The corresponding electrostatic potential (ESP) and charge difference maps were derived from these Mulliken charges to visualize the spatial distribution of charge transfer and polarization within the PI and PIS systems. This clarification has been added to the revised Supplementary Information under the computational methods section.

Supplementary Fig. 10. Frontier electron densities of surface electrostatic potential and molecular dipole for (a) PI and (b) PIS.

Supplementary Fig. 11. Frontier electron densities of *Mulliken* charge distribution on the bilayer PIS.

Supplementary Fig. 12. Frontier electron densities of *Mulliken* charge distribution on the bilayer PI.

9. In Supplementary Figure 12, I am surprised to read that there is such an asymmetric partial charge distribution in the bilayer of PIS. How is it explained?

Response to the reviewer:

The observed asymmetric partial charge distribution in the bilayer PIS arises from the incorporation of the strongly polar O=S=O groups, which induce an uneven spatial charge distribution both within and between layers. As indicated by our DFT

calculations (**Supplementary Fig. 11** and **Fig. 2a**), the upper layer of PIS exhibits significantly higher negative charge density on the O=S=O moieties than the lower layer, while the difference in the C=O group is comparatively minor. This asymmetry originates from the intrinsically polarized molecular configuration of the sulfonyl linker, which disrupts the centrosymmetric stacking observed in PI and leads to vertical electron cloud overlap between adjacent layers. Such overlap promotes the formation of “vertical conductive channels” and generates a longitudinal polarization field that facilitates interlayer carrier migration (*Angew. Chem. Int. Ed.*, 2024, **64**, e202415800; *J. Am. Chem. Soc.*, 2027, **129**, 8724–8728; *Adv. Funct. Mater.*, 2023, **33**, 2307300). Therefore, the asymmetric charge distribution is a manifestation of three-dimensional polarization and interlayer dipole coupling in the PIS bilayer, which is absent in the symmetric PI system. This anisotropic electronic configuration is essential to forming the internal electric field that drives directional charge transport and enhances photocatalytic efficiency.

Supplementary Fig. 11. Frontier electron densities of *Mulliken* charge distribution on the bilayer PIS.

Fig. 2. a The sum of charges of as-prepared samples.

10. *Supplementary Figure 35: There is an error in (a) of PIS. Text should be "average charge" instead of "average length". Supplementary Figure 35: There are 2 different types of N-O bonds. One long and three shorter. Instead of showing N-O average charges and bond-lengths, they should present the length of the long and short N-O bonds and their charge distribution as well.*

Response to the reviewer:

We have corrected the label in **Supplementary Fig. 41a**, replacing “average length” with “average charge.” In addition, we have reanalyzed the DFT calculation results to distinguish between the long and short N-O bonds in the adsorbed NO_3^- species on PI and PIS surfaces. The updated figure (**Supplementary Fig. 42**) now separately presents the bond lengths and corresponding charge distributions of the long and short N-O bonds. The results reveal that the longer N-O bond exhibits a larger charge difference, indicating preferential activation and cleavage of this bond on the PIS surface, consistent with its enhanced polarity and catalytic activity.

Supplementary Fig. 41. Comparison of (a) average charge density and (b) N-O(3) average bond length between PI- NO_3 and PIS- NO_3 .

Supplementary Fig. 42. Schematic illustrations of the NO_3 intermediates adsorbed on the surfaces of (a) PI and (b) PIS highlight the differences between the long and short N-O bonds and their corresponding oxygen atomic charges.

11. Supplementary Figure 38: How are the diffusion coefficients measured? Experimentally or computationally?

Response to the reviewer:

The diffusion coefficients presented in **Supplementary Figure 47** were obtained through molecular dynamics (MD) simulations based on DFT calculations rather than experimental measurements. Specifically, the diffusion coefficients were calculated from the mean square displacement (MSD) curves according to the Einstein relation. This computational approach allows a direct comparison of the interfacial diffusion kinetics and confirms that PIS exhibits a significantly higher NO_3^- diffusion coefficient than PI due to its enhanced surface polarity.

Mean Squared Displacement (MSD) Analysis:

The diffusion coefficient (D) was derived from the slope of the MSD curve using the Einstein relation:

$$D = \frac{1}{6N} \lim_{t \rightarrow \infty} \frac{\langle |r(t) - r(0)|^2 \rangle}{t}$$

where $r(t)$ is the position of the nitrate ion (NO_3^-) at time t , and N is the number of ions.

Linear fitting of the MSD curve (last 50 ps) provided D values.

Supplementary Fig. 47. Diffusion coefficients of PIS and PI in nitrate solution.

12. In Supplementary Figures 39 and 40, they have randomly chosen a water dimer and measured the distance. This doesn't prove their conclusions. The correct way is to plot the Radial Distribution Functionals (RDFs) from the MD snapshots between water molecules, and compare them for PI and PIS.

Response to the reviewer:

We fully agree that the radial distribution function (RDF) is a more reliable descriptor of interfacial water structure. In fact, RDF analyses have already been performed and are presented in the manuscript (**Fig. 5h**), based on MD trajectories between oxygen atoms of water molecules. The RDFs clearly show that the first-shell peak for PIS appears at 4.21 Å compared with 5.74 Å for PI, indicating a more compact and correlated water network near the PIS surface. The higher $g(r)$ value of PIS further confirms stronger hydrogen bonding and molecular confinement. **Supplementary Figs. 45-46** were included only as visual snapshots corresponding to these RDF results, to illustrate the local configuration of interfacial water. Thus, our analysis already follows the RDF-based approach suggested by the reviewer.

Fig. 5. h Radial distribution functions and coordination numbers of PIS and PI in NO_3^- .

Supplementary Fig. 45. The schematic diagram of water molecule dispersion in PI during molecular dynamics simulation.

Supplementary Fig. 46. The schematic diagram of water molecule dispersion in PIS during molecular dynamics simulation.

during molecular dynamics simulation.

Reviewer #4:

The manuscript proposes an innovative asymmetric spatial polarity strategy for donor-acceptor COFs. By integrating differentiated polar linkers, the authors achieve precise intramolecular and interlayer charge regulation, constructing polar channels and longitudinal polarization that enhance carrier migration. The optimized COF (PIS) demonstrates a high NH_4^+ production rate of $757.56 \mu\text{mol g}^{-1} \text{h}^{-1}$ and surface activity of $20362.59 \mu\text{mol m}^{-2} \text{h}^{-1}$ under natural sunlight, far exceeding most reported systems. The manuscript is well designed, comprehensive, and supported by both experimental evidence and theoretical calculations. The concept of asymmetric spatial polarity in COFs is novel and impactful, addressing key bottlenecks in photocatalytic nitrate reduction under environmentally relevant dilute conditions. However, some aspects require clarification and strengthening.

We are deeply grateful to the reviewer for their appreciation of our asymmetric spatial polarity strategy and research design. We are delighted that the novelty and significance of our approach, as well as our research findings, have been recognized. Following the suggestions, we have elaborated on relevant key points and strengthened the analyses in the revised manuscript to enhance its rigor and clarity.

1. A kinetic isotope effect (KIE) investigation using D_2O instead of H_2O is required to provide quantitative evidence of how hydrogen-bond network disruption accelerates proton transfer.

Response to the reviewer:

The kinetic isotope effect (KIE) was investigated by substituting D_2O for H_2O to probe the transfer kinetics of $^*\text{H}$ during the NO_3RR . As shown in **Fig. 5g**, the KIE values for PI and PIS, which are generally considered to be the ratios of ammonia yield/current density in H_2O and D_2O , are 1.43 and 1.16, respectively. Since all these values are greater than 1, it indicates that the dissociation of H_2O and its involvement in the hydrogenation process of intermediate nitrogen compounds constitute the rate-

determining step (RDS) for all samples. In D₂O, the NH₄⁺ yields of all catalysts exhibit a slight decline. Under identical conditions, PIS demonstrates the lowest KIE value, suggesting that the S=O=S moiety optimizes the catalytic interface, thereby facilitating the *H transfer rate. Conversely, PI exhibits the highest KIE value, indicating that the migration of *H generated from water splitting is kinetically restricted (*Appl. Catal. B- Environ.*, 2025, **360**, 124528).

Fig. 5. g KIE of H/D over PI and PIS.

2. The proposed reaction pathway is plausible. Complementary in situ techniques such as differential electrochemical mass spectrometry (DEMS) would more rigorously confirm intermediate species.

Response to the reviewer:

We fully agree that confirming the presence of reaction intermediates is essential to further validate the proposed reaction pathway. However, differential electrochemical mass spectrometry (DEMS) is mainly applicable to electrochemical systems, while our photocatalytic nitrate reduction operates in a suspended powder system, where direct coupling with DEMS is technically impractical. Instead, we propose an alternative verification strategy based on a trapping reaction for hydroxylamine (NH₂OH), the key intermediate suggested by our DFT and in situ FTIR results. Specifically, we will perform photocatalytic nitrate reduction in a 0.1 M KNO₃ solution containing cyclohexanone as a trapping reagent under visible-light irradiation. Tang *et al.* employed a catalyst to facilitate the reduction of nitrate, leading to the formation of NH₂OH. This intermediate subsequently reacts with cyclohexanone to produce cyclohexanone oxime. The latter can be readily detected using ¹H NMR spectroscopy.

(*Nat. Commun.*, 2024, **15**, 9800; *J. Am. Chem. Soc.*, 2024, **146**, 27956–27963). As shown in **Supplementary Fig. 34a**, the emergence of characteristic oxime peaks will provide direct chemical evidence for the transient formation of NH_2OH , thereby supporting the proposed stepwise reduction pathway ($^*\text{NO}_3 \rightarrow ^*\text{NO}_2 \rightarrow ^*\text{NO} \rightarrow ^*\text{NOH} \rightarrow ^*\text{NHOH} \rightarrow ^*\text{NH}_2\text{OH} \rightarrow \text{NH}_2 \rightarrow \text{NH}_3$). In addition, as shown in **Supplementary Fig. 34b**, the ^{15}N NMR spectrum of reaction solution exhibits two distinct resonances. The major peak at 376 ppm is assigned to nitrate (NO_3^-), whereas the minor signal at 610 ppm corresponds to nitrite (NO_2^-). These positions agree well with the reported ^{15}N chemical shifts of $\text{Na}^{15}\text{NO}_3$ (≈ 374 ppm) and $\text{Na}^{15}\text{NO}_2$ (≈ 617 ppm) in the literature (*Phys. Chem. Chem. Phys.*, 2023, **25**, 14538; *Chem. Commun.*, 2023, **59**, 14407). The appearance of the NO_2^- signal confirms that nitrate was partially reduced via the $\text{NO}_3^- \rightarrow \text{NO}_2^-$ intermediate pathway during the photocatalytic reaction.

Supplementary Fig. 34. (a) ^1H NMR spectrum of the reaction product. (b) The reaction product was detected by ^{15}N NMR spectra of PIS.

3. The hydrogen-bonding network disruption claim is intriguing but somewhat speculative. Stronger evidence such as direct quantification of $^\text{H}$ species would make the discussion more robust.*

Response to the reviewer:

We performed $\text{H}_2\text{O}/\text{D}_2\text{O}$ experiments, replacing H_2O with D_2O as the electrolyte solvent, to measure the kinetic isotope effect (KIE) and delve into the $^*\text{H}$ transfer kinetics during the nitrate reduction reaction (NO_3RR) (**Fig. 5g** and **Supplementary Table 6**). The KIE values for PI and PIS, typically calculated as the ratios of ammonia yield/current density in H_2O to those in D_2O , are 1.43 and 1.16, respectively. All values

above 1 reveal that H₂O dissociation and its role in hydrogenating intermediate nitrogen species are the rate-determining steps (RDS) across all samples (*Nat. Catal.*, 2019, **2**, 793–800). In D₂O, all catalysts show a minor drop in NH₄⁺ yield. Under identical conditions, PIS has the lowest KIE value, suggesting that the S=O=S group optimizes the catalytic interface, boosting *H transfer (*Appl. Catal. B-Environ.*, 2025, **360**, 124528). Conversely, PI has the highest KIE value, pointing to kinetic limitations in *H migration from water dissociation. The calculated activation free-energy differences ($\Delta\Delta G^\ddagger = RT\ln(\text{KIE})$) reinforce this: PI's $\Delta\Delta G^\ddagger$ is 0.88 kJ·mol⁻¹, almost double PIS's (0.43 kJ·mol⁻¹). PIS's lower KIE and smaller $\Delta\Delta G^\ddagger$ imply less reliance on an intact hydrogen-bonding network for proton transfer, aligning with the hypothesis of disrupted or weakened interfacial hydrogen bonds. This reduced proton-transfer sensitivity accounts for PIS's notably higher ammonia production and supports the idea that modifying the interfacial hydrogen-bonding environment enhances nitrate reduction efficiency.

Supplementary Table 6. KIE values and activation free-energy differences (298 K).

Sample	NH ₄ ⁺ rate in H ₂ O (mmol g ⁻¹ h ⁻¹)	NH ₄ ⁺ rate in D ₂ O (mmol g ⁻¹ h ⁻¹)	KIE (k_H / k_D)	$\Delta\Delta G^\ddagger$ (kJ mol ⁻¹)
PI	0.092	0.064	1.430	0.880
PIS	0.758	0.653	1.160	0.430

D₂O rates are back-calculated from measured $\text{KIE} = (\text{rate}_{\text{H}_2\text{O}} / \text{rate}_{\text{D}_2\text{O}})$.

$\Delta\Delta G^\ddagger = RT\ln(\text{KIE})$, where $R = 8.314 \text{ J} \cdot \text{mol}^{-1} \cdot \text{K}^{-1}$, $T = 298 \text{ K}$.

Fig. 5. g KIE of H/D over PI and PIS.

4. While NH_4^+ selectivity is presented to be above 90%, possible side products should be more quantitatively tracked.

Response to the reviewer:

To quantitatively track possible side products, we conducted ion chromatography (IC) analysis to detect and quantify nitrite (NO_2^-) species during the photocatalytic reaction. As shown in **Fig. 4b**, the average formation rate of NO_2^- for the PIS and PI are 1.54×10^{-3} and 1.02×10^{-3} mmol, respectively. Importantly, no N_2 or other gaseous byproducts were observed by gas chromatography. These results collectively verify that NH_4^+ is the dominant product (>90% selectivity) and that undesired side reactions are effectively suppressed during the photocatalytic process over PIS.

Fig. 4. b Selectivity of NO_3^- reduction products for as-prepared samples.

5. Post-reaction solid-state NMR or high-resolution TEM should also be provided to validate whether the asymmetric linkers preserve their chemical environment after extended operation.

Response to the reviewer:

In response, we have performed solid-state ^{13}C NMR analyses for the PIS sample before and after photocatalytic cycling to evaluate its structural stability. As shown in the newly added data (**Supplementary Fig. 30**), the ^{13}C NMR spectra displays nearly identical chemical shift positions and peak intensities, confirming that the imide and sulfonyl linkages remain intact after long-term operation. These results demonstrate that the asymmetric linker configuration and framework integrity of PIS are well maintained during repeated photocatalytic nitrate reduction, supporting the excellent durability and stability of the catalyst.

Supplementary Fig. 30. The solid-state ^{13}C NMR of PIS before and after the recycling test.

Reviewer #3 (Remarks to the Author):

The authors have addressed carefully all comments regarding the DFT calculations. In Comment 6, they have added details about the Molecular Dynamics simulations. However, the selection of the parameters for the MD simulation is poor and is not sufficiently justified.

We thank the reviewer for the thorough and critical assessment of the molecular dynamics (MD) simulations. We fully acknowledge that insufficient trajectory length, incomplete parameter justification, and missing system details may undermine the reliability of MD-based conclusions. In response to the reviewer's comments, we have substantially revised the MD simulations and the corresponding descriptions, as detailed below.

1. They have carried calculations for a duration of only 200ps=0.2 ns. This is a very short trajectory time and the accuracy of the results is questionable. Results based on trajectories less than 1 ns can not be trusted. The minimum trajectory time, accepted for publications, is 10 ns.

Response to the reviewer: We agree with the reviewer that a trajectory length of 200 ps is insufficient to ensure reliable statistical sampling for MD analyses. Following the reviewer's recommendation, we have completely re-performed all MD simulations with an extended total simulation time of 10 ns, which meets the commonly accepted standard for MD studies involving interfacial dynamics and diffusion behavior. The MD time step was set to 1.0 fs, ensuring numerical stability. All analyses reported in the revised manuscript are based exclusively on the equilibrated production trajectories. The original 200 ps results have been fully replaced by the new 10 ns simulations (**Supplementary Fig. 47 and Fig. 5h**).

Supplementary Fig. 47. Diffusion coefficients of PIS and PI in nitrate solution.

Fig. 5. h Radial distribution functions and coordination numbers of PIS and PI in NO_3^- .

2. Moreover, convergence of the results (e.g energy, pressure, diffusion coefficients) should be checked with respect to the simulation time.

Response to the reviewer: All molecular dynamics (MD) simulations were performed in the canonical (NVT) ensemble, employing a Nosé-Hoover thermostat to maintain the system temperature at 298 K. Throughout the production run, time-dependent profiles of total energy were continuously monitored. The results indicate that the system achieved a stable equilibrium state, as evidenced by a drift of less than 0.1% in the aforementioned physical quantity over a 10 ns simulation period. Mean square displacement (MSD) curves were evaluated over the full production trajectory. Diffusion coefficients calculated using block averaging over independent time windows (0-1 ns, 1-2 ns, 2-3 ns, 3-4 ns, 4-5 ns, 5-6 ns, 6-7 ns, 7-8 ns, 8-9 ns, 9-10 ns), showing deviations within $\pm 5\%$, indicating good convergence with respect to simulation time. These results demonstrate that the MD observables reported in this work are statistically converged (**Fig. 1**).

Fig. 1 a Continuously monitor the time-varying profile of total energy throughout the entire production cycle. **b** The relative error of the diffusion coefficient for each block in PIS.

3. In the last sentence, they mention "These parameters and protocols are consistent with those employed in similar studies on COF-water interfacial dynamics and nitrate diffusion behavior (Chem. Eng. J., 2022, 441, 136084; Appl. Catal. B, 2025, 361, 124558)." These 2 articles are previous works from some of the authors of the current manuscript. However, in these 2 articles only DFT calculations have been utilized, and no MD simulations. So, these MD parameters haven't been used before to describe dynamics of COF-Water systems.

Response to the reviewer: We thank the reviewer for pointing out this error. We acknowledge that the cited studies employed static DFT calculations rather than MD simulations. The corresponding statement has therefore been removed from the revised manuscript. The MD parameters and protocols used in the present work are now clearly described as being newly implemented in this study, based on standard MD practices rather than prior publications.

4. Moreover, there is no explanation about how many water molecules are added in the simulation box. Therefore, MD results are still questionable and cannot be trusted and published. Either the MD results and conclusions are omitted, or the authors need to perform MD simulations with the minimum required settings.

Response to the reviewer: All MD simulations were carried out using the Forcite Plus module in Materials Studio 2020. The calculations employed a classical force field approach (COMPASS III force field), which accurately describes both organic

frameworks and solvent interactions. The simulations were performed in the NVT ensemble (constant number of particles, volume, and temperature) using a Nosé-Hoover thermostat to maintain the temperature at 298 K. The integration time step was set to 1.0 fs, and the total simulation time was 10 ns. The simulation system employed periodic boundary conditions (PBCs) in all three spatial dimensions. The cutoff radius for nonbond interactions was set to 12 Å, and long-range electrostatic interactions were treated using the particle mesh Ewald (PME) method with an error tolerance $\leq 10^{-5}$, ensuring an accurate description of ion-water and ion-ion interactions. Snapshots were saved every 1 ps during the 10 ns production run, resulting in approximately 10000 configurations. The final simulation cell contains 100 water molecules and 5 nitrate ions. In response to the reviewer's concerns, we have chosen the latter option and comprehensively upgraded the MD simulations. With the extended 10 ns trajectories, explicit convergence analyses, corrected literature citations, and fully specified system setup, we believe that the MD results now meet the minimum and commonly accepted requirements for publication.

Reviewer #3 (Remarks to the Author):

The authors have performed additional MD simulations and addressed carefully all comments regarding the MDs. I have 4 minor corrections:

We thank the reviewer for the positive evaluation. All MD-related comments have been carefully addressed by refining the analysis and clarifying the description in the revised manuscript. The four minor corrections have been fully incorporated.

1. Supplementary Material, Page 2, L32-33: "To eliminate the van der Waals forces" should be changed to "To account for the van der Waals interactions"

Response to the reviewer: We thank the reviewer for pointing this out. The wording has been corrected to “To account for the van der Waals interactions” in the revised Supplementary Material (Page 2, Lines 32-33).

2. Supplementary Material, Page 2, L33, "we adopted Grimme methods for DFT-D correction": The authors should mention which one of the Grimme's method for dispersion corrections. While in their previous response letter (page 21), they mention which DFT-D was used, they haven't updated the Supplementary Material. A reference of the DFT-D method should also be given.

Response to the reviewer: We thank the reviewer for this helpful comment. The use of Grimme’s DFT-D3 method with Becke-Johnson damping (DFT-D3(BJ)) (*J. Comput. Chem.* 2011, **32**, 1211–1216) for dispersion correction has been explicitly described in the Supplementary Material (Page 2, Line 33), and the corresponding reference has been included accordingly (*Corros. Sci.* 2024, **236**, 112237).

3. Supplementary Material, Page 3, L73-77: Did they consider all configurations from the 10ns trajectory for the RDF graphs, or only a last section of them? Usually, the first few ns of the trajectory are not taken into account for the calculation of properties, and only the last 2-3 ns are considered. This is done in order to ensure that the system has reached equilibration.

Response to the reviewer: We thank the reviewer for this important point. The first part of the MD trajectory was treated as the equilibration stage (2 ns) and was not used for property analysis. All RDFs and related statistical quantities were calculated using only the equilibrated portion of the trajectory, specifically the last 3 ns of the 10 ns

simulation. This clarification has now been explicitly added to the Supplementary Material (Page 3).

4. Article, Page 19, L459-463: The authors report Radial Distribution Graphs in Fig 5h, but they don't mention which pair is taken into account.

Response to the reviewer: We thank the reviewer for this insightful comment. In Fig. 5h, the RDFs were calculated between specific atoms of the nitrate species and the defined active sites on the catalyst surfaces. For the PIS system, the RDF corresponds to the distribution between the oxygen atoms of nitrate ions and the sulfonyl oxygen atoms (O=S=O) of PIS, which serve as the dominant adsorption and interaction sites. In contrast, for the PI system, the RDF was calculated between the oxygen atoms of nitrate ions and the carbonyl oxygen atoms (C=O) of PI, which act as the primary adsorption sites.

To avoid ambiguity, we have now explicitly clarified these RDF atom pairs and their corresponding active sites in the main text (Page 19). The revised text now reads:

“MD simulations were carried out to analyze the atomic structures of interfacial H₂O on various material surfaces. As illustrated in **Fig. 5h** and **Supplementary Figs. 45-46**, radial distribution function (RDF) analysis of the H₂O-NO₃⁻ system, calculated between nitrate oxygen atoms and the surface active sites (O=S=O in PIS and C=O in PI), reveals that the first-shell peak of interfacial water molecules appears at 5.22 Å for PIS, which is shorter than that of PI (5.60 Å), indicating a more compact and ordered water network on the PIS surface.”